# Global Convergence and Rich Feature Learning in $L$-Layer Infinite-Width Neural Networks under $\mu$ Parametrization

**Zixiang Chen** [* 1]   **Greg Yang** [* 2]   **Qingyue Zhao** [1]   **Quanquan Gu** [1]

## Abstract

Despite deep neural networks' powerful representation learning capabilities, theoretical understanding of how networks can simultaneously achieve meaningful feature learning and global convergence remains elusive. Existing approaches like the neural tangent kernel (NTK) are limited because features stay close to their initialization in this parametrization, leaving open questions about feature properties during substantial evolution. In this paper, we investigate the training dynamics of infinitely wide, $L$-layer neural networks using the tensor program (TP) framework. Specifically, we show that, when trained with stochastic gradient descent (SGD) under the Maximal Update parametrization ($\mu$P) and mild conditions on the activation function, SGD enables these networks to learn linearly independent features that substantially deviate from their initial values. This rich feature space captures relevant data information and ensures that any convergent point of the training process is a global minimum. Our analysis leverages both the interactions among features across layers and the properties of Gaussian random variables, providing new insights into deep representation learning. We further validate our theoretical findings through experiments on real-world datasets.

## 1. Introduction

Deep learning has achieved remarkable success in various machine learning tasks, from image classification (Krizhevsky et al., 2012) and speech recognition (Hinton et al., 2012) to game playing (Silver et al., 2016). Yet this empirical success has posed a significant theoretical challenge: how can we explain the effectiveness of neural networks given their non-convex optimization landscape and over-parameterized nature? Traditional optimization and learning theory frameworks struggle to provide satisfactory explanations. A breakthrough came with the study of infinite-width neural networks, where the network behavior can be precisely characterized in the limit of infinite width. This theoretical framework has spawned several important approaches to understanding neural networks, with the Neural Tangent Kernel (NTK) emerging as a prominent example.

Under the NTK parametrization (NTP) (Jacot et al., 2018), neural network training behaves like a linear model: the features learned during training in each layer remain essentially identical to those obtained from random initialization. Consequently, the training process of over-parameterized deep neural networks can be characterized by training linear models with random feature (Lee et al., 2019; Arora et al., 2019b; Cao & Gu, 2019; Chen et al., 2021). Since random features are linearly independent, global convergence can be proved for wide neural networks trained using (stochastic) gradient descent (GD/SGD) (Du et al., 2019c; Allen-Zhu et al., 2019; Du et al., 2019a; Zou et al., 2019; Zou & Gu, 2019). However, the NTK parametrization has significant limitations, such as its inability to perform feature learning and transfer learning, which involve pretraining and fine-tuning. While NTK theory provides convergence results under infinite width, its inability to explain feature learning motivates us to ask:

*Can deep neural networks simultaneously learn meaningful features and achieve global convergence?*

In this paper, we show that deep neural networks can achieve both objectives through proper parametrization. While previous approaches like NTK and standard parametrization fail to perform meaningful feature learning, and mean field parametrization suffers from feature collapse in deep networks, we demonstrate that the $\mu$ parametrization (Yang & Hu, 2020; 2021; Yang et al., 2021; Yang, 2019a) enables both feature learning and global convergence. Specifically, working with $L$-layer neural networks under $\mu$P scaling, we prove that despite substantial feature evolution during train-

---

[*]Equal contribution [1]Department of Computer Science, University of California, Los Angeles, USA [2]xAI, USA. Correspondence to: Quanquan Gu <qgu@cs.ucla.edu>.

*Proceedings of the 42nd International Conference on Machine Learning*, Vancouver, Canada. PMLR 267, 2025. Copyright 2025 by the author(s).

ing, the networks maintain linearly independent features in each layer when trained with stochastic gradient descent. As a consequence, if the training converges, it must converge to a global minimum. Our contributions are summarized as follows:

- We establish that multilayer perceptrons (MLPs) under Maximal Update Parametrization ($\mu$P) learn linearly independent features that capture task-relevant information. The learned features substantially deviate from their initialization, demonstrating true feature learning rather than random feature approximation. This resolves a fundamental challenge in deep learning theory: characterizing feature properties that ensure global convergence while allowing meaningful feature learning.

- Our proof technique analyzes neural network Gaussian processes by exploiting their second-order invariants across adjacent layers. These structural properties persist during training, which allows us to track the evolution of feature correlations. Through a careful inductive argument over network layers and iterations, we establish that when training converges, the linear independence of features ensures convergence to a global minimum. The proof reveals a deep connection between the feature learning dynamics and the structural properties of infinite-width neural networks.

- Through experiments on classification tasks, we validate our theoretical findings by demonstrating that features maintain linear independence through analysis of covariance matrix properties. Our empirical results demonstrate $\mu$P's unique capability to simultaneously achieve meaningful feature learning while preserving feature richness, as supported by non-vanishing eigenvalues as network widths increase. Through comparative analysis against other parametrization schemes, we show that this behavior robustly persists across different choices of activation functions, illustrating the practical implications of our theoretical results.

**Notation.** For any positive integer $N$, we use $[N]$ to denote the index set $\{1, \ldots, N\}$. We use $\phi : \mathbb{R} \to \mathbb{R}$ to denote the activation function. For an $L$-layer network, we use superscript $l \in [L]$ to index layers, with $Z^{h^l}$ and $Z^{x^l}$ denoting pre-activation and post-activation features respectively.

For matrices and vectors, $\widehat{W}_0^{L+1} := W_0^{L+1} n$ denotes a scaled last layer weights. For any matrix $W$ and vector $x$, $\widehat{Z}^{Wx}$ denotes the Gaussian component of $Z^{Wx}$.[1] We use $\mathbb{E}[\cdot]$ to denote expectation.

We consider a filtration $\{\mathcal{F}_t\}_{t \geq 0}$, where $\mathcal{F}_t$ is the $\sigma$-algebra generated by all random variables up to time $t$. This gives a

---

[1] $Z^{Wx} := \widehat{Z}^{Wx} + \dot{Z}^{Wx}$, which is detailed in Appendix B.

sequence of probability spaces $(\Omega, \mathcal{F}_t, \mathbb{P})$ with $\mathcal{F}_0 \subseteq \mathcal{F}_1 \subseteq \ldots \subseteq \mathcal{F}_T$. An event $\mathcal{E} \in \mathcal{F}_T$ occurs almost surely (denoted as a.s.) if $\mathbb{P}(\mathcal{E}) = 1$. The functions $\mathring{f}$ and $\mathring{\chi}$ denote the infinite-width limits of network outputs and error signals induced by $\mathring{f}$ respectively.

## 2. Related Work

**Neural Tangent Kernel Parametrization** Jacot et al. (2018) first introduced the neural tangent kernel (NTK) by studying the training dynamics of multi-layer perceptrons (MLPs) with Lipschitz and smooth activation functions under square loss. Based on NTK, Allen-Zhu et al. (2019); Du et al. (2019a); Zou et al. (2019); Arora et al. (2019a) proved the global convergence of (stochastic) gradient descent for various neural architectures with general activation and loss functions. Standard parametrization (SP) and NTK parametrization (NTP) share the same weight initialization scheme but with different learning schedules. As network width increases, SP requires learning rates to decrease as $O(1/\text{width})$ for all layers to maintain stability (Yang & Hu, 2020). When considering the infinite-width limit, neither SP nor NTK parametrization can learn features - the features remain essentially the same as those from random initialization. Both theoretical studies and empirical evidence demonstrated that these parametrizations failed to capture the feature learning behavior observed in practical neural networks (Woodworth et al., 2020; Geiger et al., 2020; Bordelon & Pehlevan, 2022; Yang et al., 2023a). Everett et al. (2024) shows that SP and NTP can empirically exhibit feature learning when using large per-layer learning rate exponents. However, the existing NTK analysis framework cannot directly analyze feature learning in SP and NTP under this setting, leaving open questions about convergence guarantees during such feature learning.

**Mean Field Analysis** The mean field limit emerged when networks and learning rates were scaled appropriately as width approached infinity, yielding nonlinear parameter evolution (Mei et al., 2018; Chizat & Bach, 2018; Rotskoff & Vanden-Eijnden, 2018; Sirignano & Spiliopoulos, 2018). Early analysis of two-layer networks showed promising results, proving convergence to global optima with explicit convergence rates established through both direct analysis (Chen et al., 2020) and mean field Langevin dynamics (Nitanda et al., 2022). Progress extended to three-layer networks with Pham & Nguyen (2021)'s global convergence results. However, studies of deeper architectures revealed significant limitations: for networks deeper than 4 layers, both feature vectors and gradients degenerated to zero vectors (Nguyen & Pham, 2020; Fang et al., 2021). While Hajjar et al. (2021) introduced Integrable Parameterization (IP) to address this, networks with more than four layers still started at a stationary point in the infinite-width limit,

hard to achieve rich feature learning.

**Tensor Programs** Tensor Programs (TPs) emerged as a unified framework for understanding infinite-width limits across neural architectures (Yang, 2019b; 2020a;b). This approach generalized previous architecture-specific parametrizations (Du et al., 2018; 2019b; Hron et al., 2020; Alemohammad et al., 2020). Yang & Hu (2020) characterized two distinct behaviors in infinite-width MLPs: one where initialization dominated the training dynamics (the kernel regime), and another where training data substantially influenced the learned weights (the feature learning regime). Within this framework, the $\mu$ parametrization was identified as enabling maximal feature learning across all layers and architectures (Yang & Hu, 2020; Yang et al., 2021; Littwin & Yang, 2022). The framework has continued to expand with analysis of depth-dependent scaling (Yang et al., 2023c). Recent work by Yang et al. (2023a) refined the understanding through spectral analysis and input dimension scaling, which we adopt in our experiments.

Our experimental results reveal distinct feature learning behaviors across different parametrization schemes. As shown in Figure 1, Standard Parametrization (SP) keeps features close to initialization (demonstrated by small feature change in the left panel), while Integrable Parametrization (IP) achieves feature learning but suffers from feature collapse (shown by decreasing feature diversity in the right panel). In contrast, $\mu$P achieves both substantial feature change and maintains feature diversity. We summarize these key characteristics in Table 1. Additional experiments with different activation functions, further illustrating these trends, are provided in Appendix A.

*Table 1.* Feature Properties Under Different Parametrizations

| Parametrization | Feature Learning | Feature Richness |
|---|---|---|
| Standard (SP) | ✗ | Rich |
| Neural Tangent (NTP) | ✗ | Rich |
| Meanfield (IP)[2] | ✓ | Low |
| Maximal Update ($\mu$P) | ✓ | Rich |

## 3. Preliminaries

Different parametrization schemes for MLPs are shown in Table 2 [3]. Given a general MLP with $L$ hidden layers specified by weight matrices $W^1 \in \mathbb{R}^{n \times d}$, $\{W^l\}_{l=2}^L \in \mathbb{R}^{n \times n}$, $W^{L+1} \in \mathbb{R}^n$, and activation $\phi : \mathbb{R} \to \mathbb{R}$, the network

---

[2]IP (Integrable Parametrization) refers to parametrizations with a $1/n$ scaling factor for all layers except the first one, which leads to absolute convergence of weighted sums in the mean-field limit.

[3]Init. Var. denotes initialization variance, LR denotes learning rate scaling. $\eta$ is the base learning rate and $n$ is the layer width. For notational simplicity, we omit the constant in the table.

computation is formally defined as

$$
\begin{aligned}
h^1 &= W^1 \xi \in \mathbb{R}^n, \\
x^l &= \phi(h^l) \in \mathbb{R}^n, \\
h^{l+1} &= W^{l+1} x^l \in \mathbb{R}^n, \\
f(\xi) &= W^{L+1} x^L \in \mathbb{R}, \quad (3.1)
\end{aligned}
$$

where $L > 1$ is any positive integer and $l \in \{1, \ldots, L-1\}$. Among these schemes, the *Maximal Update Parametrization* ($\mu$P) shown in Table 2 achieves maximal parameter updates at initialization. As $n \to \infty$, we can consider the following infinite-width feature learning process: $f_t(\xi) \stackrel{a.s.}{\to} \mathring{f}_t(\xi)$ (Yang & Hu, 2020, Theorem 6.4). The neural network is assumed to be trained using a differentiable loss $\mathcal{L}$ by stochastic gradient descent, where the $s$-th sampled batch is denoted by $\{(\xi_i, y_i)\}_{i \in \mathcal{B}s} \subseteq S$ where $\mathcal{B}_s$ is the index set and $S$ is the training dataset. For simplicity, we present the full-batch gradient descent result in the main paper, i.e., $\mathcal{B}_s = |S| = [m]$.

**Represent Hidden States via $Z$ Random Variables:** Following Yang & Hu (2020), we represent network's hidden states using $Z$ random variables. This representation generalizes the spirit of two-layer mean field analysis: even with multiple hidden layers ($L \geq 2$), the entries of preactivation $h$ and activation vectors $x$ in (3.1) become approximately i.i.d. as width $n$ approaches infinity. This allows us to characterize their asymptotic behavior using scalar random variables that reflect their elementwise distributions.

Specifically, for a vector $x \in \mathbb{R}^n$, we track it using $Z^x$, where $x$'s entries behave like i.i.d. copies of $Z^x$. When $x$ is properly scaled such that $\|x\|_2^2 = \Theta(n)$ (i.e., its typical magnitude is independent of $n$), then $Z^x$ becomes independent of $n$. For any two such normalized vectors $x, y \in \mathbb{R}^n$, their corresponding random variables $Z^x$ and $Z^y$ are correlated via $\lim_{n \to \infty} x^\top y / n = \mathbb{E} Z^x Z^y$. Our goal is to characterize these $Z$ in (3.1) throughout the training process.

**Definition 3.1.** [Yang & Hu 2020] During training, we define the error signal $\mathring{\chi}_{t,i}$ at time step $t$ for the $i$-th sample. When training with SGD to minimize the loss function $\mathcal{L}$, this error signal is computed as $\mathring{\chi}_{t,i} = \mathcal{L}'(\mathring{f}_t, y_i) \mathbb{1}\{i \in \mathcal{B}_t\}$, where $\mathring{f}_t$ is the model output at time $t$, $(\xi_i, y_i)$ is the $i$-th training sample pair, and $\mathcal{B}_t$ denotes the mini-batch at time step $t$. The indicator function $\mathbb{1}\{\cdot\}$ ensures that the error signal is only computed for samples in the current mini-batch.

This error signal captures how much the model's prediction deviates from the true label for each sample in the current mini-batch, and serves as the driving force for parameter updates during SGD training. For instance, in the case of mean squared error loss, the error signal takes the form $\mathring{\chi}_{t,i} = 2(\mathring{f}_t(\xi_i) - y_i) \mathbb{1}\{i \in \mathcal{B}_t\}$. Having defined the error signal, we now describe how the Z-variables characterize the

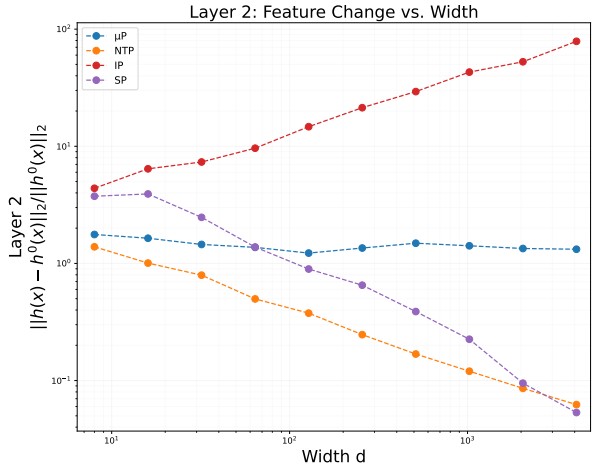 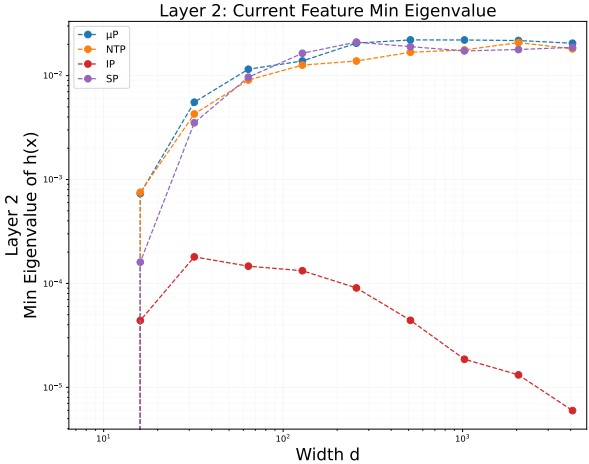

*Figure 1.* Different parametrization schemes exhibit distinct feature learning behaviors as width increases in 3 hidden-layer MLPs. We train it on the CIFAR-10 dataset and measure pre-activation feature properties in the second hidden layer (Layer 2). Left: Feature change ($\|h(x) - h^0(x)\|_2/\|h^0(x)\|_2$) shows only $\mu$P maintain stable feature representations. Right: Feature diversity measured by the minimum eigenvalue of the feature gram matrix $K_{ij} = \langle h(x_i), h(x_j) \rangle$, where a larger eigenvalue indicates the features span a higher dimensional space. The results reveal that Meanfield parametrization suffers from feature collapse while SP, NTP and $\mu$P preserve rich feature representations. Notably, only $\mu$P achieves both feature learning capability and feature richness. See Appendix A for experimental details.

*Table 2.* Initialization variance and learning rate scaling under different parametrization schemes for MLP networks.

| **Layer** | **SP** | | **NTP** | | **IP** | | **$\mu$P** | |
|---|---|---|---|---|---|---|---|---|
| | Init. Var. | LR | Init. Var. | LR | Init. Var. | LR | Init. Var. | LR |
| Input ($W^1$) | 1 | $\eta \cdot n^{-1}$ | 1 | $\eta$ | 1 | $\eta \cdot n$ | 1 | $\eta \cdot n$ |
| Hidden ($W^l$) | $n^{-1}$ | $\eta \cdot n^{-1}$ | $n^{-1}$ | $\eta \cdot n^{-1}$ | $n^{-2}$ | $\eta$ | $n^{-1}$ | $\eta$ |
| Output ($W^{L+1}$) | $n^{-1}$ | $\eta \cdot n^{-1}$ | $n^{-1}$ | $\eta \cdot n^{-1}$ | $n^{-2}$ | $\eta \cdot n^{-1}$ | $n^{-2}$ | $\eta \cdot n^{-1}$ |

network's computation in the infinite-width limit $\mathring{f}_t(\xi)$. The forward pass tracks how network features propagate through layers, while the backward pass characterizes gradient flow. For clarity of presentation, we next introduce a simplified version of $\mathring{f}$ that includes the key properties needed for our theoretical analysis. The complete derivation and technical details can be found in Appendix B.

**Forward Pass**

1. For $z \in \{x^l, h^l\}_l$, we have $Z^{z_t(\xi)} = Z^{z_0(\xi)} + Z^{\delta z_1(\xi)} + \cdots + Z^{\delta z_t(\xi)}$ where

   (a) for $l \in [L]$, $Z^{\delta x_t^l(\xi)} = \phi(Z^{h_t^l(\xi)}) - \phi(Z^{h_{t-1}^l(\xi)})$,
   (b) for $l = 1$, we have
   $$Z^{\delta h_t^l(\xi)} = -\sum_{i \in [m]} \eta \mathring{\chi}_{t-1,i} \xi_i^\top \xi Z^{dh_{t-1}^l(\xi_i)},$$

   for $2 \le l \le L$, we have
   $$Z^{\delta h_t^l(\xi)} = \widehat{Z}^{W_0^l \delta x_t^{l-1}(\xi)} + F_t(\xi), \qquad (3.2)$$

   where $F_t$ is a function that is determined by the random variable $\{Z^{dh_s(\xi_i)}\}_{i \in [m], s \in [t-1]}$ (see Ap-

pendix B for detail), and $\widehat{Z}^{W_0^l \delta x_t^{l-1}(\xi)}$ are zero centered jointly Gaussian with covariance matrix

$$\text{Cov}(\widehat{Z}^{W_0^l \delta x_t^{l-1}(\xi)}, \widehat{Z}^{W_0^l \delta x_s^{l-1}(\varsigma)}) = \mathbb{E}[Z^{\delta x_t^{l-1}(\xi)} Z^{\delta x_s^{l-1}(\varsigma)}].$$

2. For last layer weight, we have $Z^{\widehat{W}_t^{L+1}} = Z^{\widehat{W}_0^{L+1}} + Z^{\delta W_1^{L+1}} + \cdots + Z^{\delta W_t^{L+1}}$ where
   $$Z^{\delta W_t^{L+1}} = -\eta \sum_{i \in [m]} \mathring{\chi}_{t-1,i} Z^{x_{t-1}^L(\xi_i)}. \qquad (3.3)$$

3. The output deltas have limits $\mathring{f}_t(\xi) = \delta \mathring{f}_1(\xi) + \cdots + \delta \mathring{f}_t(\xi)$ where
   $$\delta \mathring{f}_t(\xi) = \mathbb{E} Z^{\delta W_t^{L+1}} Z^{x_t^L(\xi)} + \mathbb{E} Z^{\widehat{W}_{t-1}^{L+1}} Z^{\delta x_t^L(\xi)}. \qquad (3.4)$$

**Backward Pass**

1. For gradients:
   $$Z^{dx_t^L(\xi)} = Z^{\widehat{W}_t^{L+1}} \qquad (3.5)$$
   $$Z^{dh_t^l(\xi)} = Z^{dx_t^l(\xi)} \phi'(Z^{h_t^l(\xi)}) \qquad (3.6)$$
   $$Z^{dx_t^{l-1}(\xi)} = \widehat{Z}^{W_0^{l\top} dh_t^l(\xi)} + G_t(\xi) \qquad (3.7)$$

where $G_t$ is function that is determined by the random variable $\{Z^{x_s^{l-1}(\xi_i)}\}_{i\in[m],s\in[t-1]}$ (see Appendix B for detail), and where $\{\widehat{Z}^{W_0^{l\top} dh_t^l(\xi)}\}_{\xi,t}$ are zero centered jointly Gaussian with covariance matrix

$$\text{Cov}(\widehat{Z}^{W_0^{l\top} dh_t^l(\xi)}, \widehat{Z}^{W_0^{l\top} dh_s^l(\boldsymbol{\varsigma})}) = \mathbb{E}[Z^{dh_t^l(\xi)} Z^{dh_s^l(\boldsymbol{\varsigma})}].$$

**Remark 3.2.** The error signal generalizes to different optimization objectives. For binary classification problems, it can be expressed as $\mathring{\chi}_{t,i} = -y_i/\big(1 + \exp(y_i \cdot \mathring{f}_t(\xi_i))\big)$.[4]

## 4. Main Results

In this section, we present our main theoretical results, which rely on the following assumptions regarding the training data and activation function. Specifically, we will first state a mild geometric condition on the inputs, and then discuss the regularity requirements on the activation function.

**Assumption 4.1.** Consider input vectors $\xi$ drawn from the training data set $S \subseteq \mathbb{R}^d$ satisfying that for any three different points $\xi_i, \xi_j, \xi_k \in S$, the following property holds,

$$|\langle \xi_i, \xi_j \rangle| \neq |\langle \xi_i, \xi_k \rangle|, \quad |\langle \xi_i, \xi_j \rangle| \neq 0, \forall i \neq j.$$

Assumption 4.1 rules out the possibility of identical or zero inner products among different data points, which could otherwise lead to degenerate analyses. Although it may appear restrictive, it holds with probability 1 if the samples are drawn from any continuous distribution (e.g., Gaussian). Indeed, the set of points violating the above requirement—such as those with exactly matching inner products—has Lebesgue measure zero. In practice, minor random perturbations to discrete data can also ensure the condition is satisfied.

**Definition 4.2** (GOOD Function). A function $\phi : \mathbb{R} \to \mathbb{R}$ is called GOOD if it prevents degeneracy in neural networks by ensuring non-trivial compositions. Specifically, for any finite set of parameters $\{a_i\}, \{b_i\}, \{c_i\}$ satisfying $a_k b_k \neq 0, \exists k$ and $|b_i| \neq |b_j|, \forall i \neq j$ we have that the composite mapping

$$f(x) = \sum_{i=1}^{n} a_i \phi(b_i x + c_i), \quad x \in \mathbb{R}$$

is not a constant function. Moreover, for any real numbers $r_1, r_2$, the function $(r_1 + \phi(x))(r_2 + \phi'(x))$ is not almost everywhere constant.

We next introduce an assumption on the activation function that ensures it is both sufficiently smooth and GOOD:

**Assumption 4.3.** We assume that the activation function $\phi$ satisfies the following properties.

1. $\phi$ is twice continuously differentiable.

2. $\phi'$ and $\phi''$ are bounded.

3. $\phi$ is a GOOD function.

4. $\{x \in \mathbb{R} : \phi(x) = y\}$ and $\{x \in \mathbb{R} : \phi'(x) = y\}$ are countable for all $y \in \mathbb{R}$.

**Remark 4.4.** Assumption 4.3 imposes regularity and smoothness conditions on the activation function, ensuring that $\phi'$ is pseudo-Lipschitz[5], a requirement for Yang & Hu (2020, Theorem 7.4). These conditions are met by many commonly used activation functions, including the sigmoid function $\sigma(x) = 1/\big(1 + \exp(-x)\big)$ and hyperbolic tangent (tanh), which is a rescaled version of sigmoid.

Modern activation functions such as the SiLU (Sigmoid Linear Unit), defined as $\text{SiLU}(x) = x \cdot \sigma(x)$ (Hendrycks & Gimpel, 2016), also satisfy these assumptions. SiLU has been widely adopted in practice, including in several state-of-the-art open-source foundation models (Touvron et al., 2023a;b). A detailed discussion of activation functions that meet these criteria is provided in Appendix D.

With these assumptions in place, we can now state our main theoretical results regarding feature non-degeneracy and convergence. In particular, the following theorem establishes that in wide neural networks, feature representations evolve while maintaining their diversity and avoiding collapse throughout training.

**Theorem 4.5.** Consider an infinite-width $L$-layer MLP trained with gradient descent. Under Assumptions 4.1 and 4.3, the features in each layer are non-degenerate at any time $t$ during training. Specifically, for each layer $l \in [L]$:

1. The pre-activation features $\{Z^{h_t^l(\xi)}\}_{\xi \in S}$ are linearly independent.

2. The post-activation features $\{Z^{x_t^l(\xi)}\}_{\xi \in S}$ are linearly independent.

This non-degeneracy property has important implications for the convergence behavior of the model. In particular, it allows us to characterize the state of the model at convergence, as described in the following corollary.

**Corollary 4.6.** Consider an infinite-width $L$-layer MLP under the conditions of Theorem 4.5. If the model converges at time $T$, meaning that the model weights remain unchanged

---

[4]$\mathcal{L}$ is only required to be continuously differentiable with respect to its first argument (Yang & Hu, 2020), which we omit in subsequent presentations.

[5]See Yang & Hu (2020, Definition E.3) for the definition of pseudo-Lipschitz functions.

for all $t \geq T$, then the error signal vanishes for all subsequent mini-batches:

$$\mathring{\chi}_{T,i} = 0, \quad \forall i \in \bigcup_{t \geq T} \mathcal{B}_t,$$

where $\mathcal{B}_t$ denotes the mini-batch at time $t$.

This corollary establishes that feature non-degeneracy forces convergence to occur only at critical points where the error signal vanishes. More precisely, when the network converges, the error signals must vanish across all samples in subsequent mini-batches, implying convergence to a global minimum of the training objective. This is a consequence of the feature non-degeneracy established in Theorem 4.5, as non-degenerate features ensure that weight updates can only stop when the network has effectively minimized the error signals.

## 5. Key Techniques and Analysis

In this section, we first identify the key technical challenges in establishing our main results, and then present the techniques and insights to address them. We begin by discussing two fundamental challenges: the tension between feature evolution and Structural stability, and the intricate coupling across network layers. We then develop a systematic framework based on Gaussian processes to overcome these challenges. The complete proof is presented in Appendix C.

### 5.1. Technical Challenges

Establishing global convergence while allowing meaningful feature learning presents two fundamental technical challenges that must be addressed simultaneously:

1. **Feature Evolution vs. Structural Stability:** In contrast to the NTK parameterization (where features stay near their initialization), $\mu$P enables features to evolve substantially during training. Specifically, for any feature $z \in \{x^l, h^l\}_l$ in the Forward Pass (a) of Section 3, we have:

$$Z^{z_t(\xi)} = Z^{z_0(\xi)} + \underbrace{Z^{\delta z_1(\xi)} + \cdots + Z^{\delta z_t(\xi)}}_{\text{feature learning term}}.$$

The presence of the feature learning term makes it challenging to track and characterize features' properties throughout optimization. This contrasts sharply with the setting under NTK parametrization, where $Z^{z_t(\xi)}$ stays equal to its initialization $Z^{z_0(\xi)}$ (Yang & Hu, 2020) - a mathematically simpler but limited case where the network behavior is fully determined by the initial kernel.

2. **Cross-Layer Coupling:** In deep networks, changes in one layer's features affect both earlier and later layers

through forward and backward propagation. For forward propagation in layer $l$ and backward propagation in layer $l + 1$, we have by (3.2) and (3.7):

$$Z^{\delta h_t^l(\xi)} = \widehat{Z}^{W_0^l \delta x_t^{l-1}(\xi)} + F_t(\xi) \tag{5.1}$$

$$Z^{dx_s^l(\xi)} = \widehat{Z}^{W_0^{l+1\top} dh_s^{l+1}(\xi)} + G_s(\xi), \tag{5.2}$$

where $F_t$ and $G_s$ capture the historical dependencies through previous features $\{Z^{dh_s^{l-1}(\xi_i)}\}_{i \in [m], s \in [t-1]}$ and gradients $\{Z^{x_s^{l-1}(\xi_i)}\}_{i \in [m], s \in [t-1]}$ respectively. This intricate coupling between forward and backward passes makes it challenging to ensure that features remain well-behaved as they propagate through the network.

Our key insight in addressing these challenges lies in analyzing structural invariants preserved by the induced Gaussian processes during training. While features evolve substantially, we find that certain second-order properties—specifically, non-degeneracy—remain invariant across layers and time steps. This invariance ensures rich feature learning while preventing the network from getting stuck in local minima.

### 5.2. The Gaussian Process View

In the infinite-width limit, neural network training induces two families of Gaussian processes that capture forward and backward propagation:

$$\{\widehat{Z}^{W_0^l \delta x_s^{l-1}(\xi_i)}\}_{i \in [m], s \in [t], 2 \leq l \leq L}, \tag{5.3}$$

$$\{\widehat{Z}^{W_0^{l\top} dh_s^l(\xi_i)}\}_{i \in [m], s \in [t], 2 \leq l \leq L}. \tag{5.4}$$

The forward process ((5.3)) tracks how features evolve across layers, while the backward process ((5.4)) describes gradient flow. Unlike prior work that studies these processes in isolation, we discover fundamental connections between their structural properties that enable both feature learning and convergence.

**Covariance Structure of Gaussian Processes** Our key technical insight is that these Gaussian processes (5.3) and (5.4) exhibit invariant covariance properties that persist throughout training. Recall from (5.1) and (5.2) that both forward and backward propagation can be decomposed into a Gaussian term and a history-dependent term:

$$Z^{\delta h_t^l(\xi)} = \underbrace{\widehat{Z}^{W_0^l \delta x_t^{l-1}(\xi)}}_{\text{Gaussian term}} + \underbrace{F_t(\xi)}_{\text{history term}}$$

$$Z^{dx_s^l(\xi)} = \underbrace{\widehat{Z}^{W_0^{l+1\top} dh_s^{l+1}(\xi)}}_{\text{Gaussian term}} + \underbrace{G_s(\xi)}_{\text{history term}}$$

We notice that these Gaussian terms can preserve covariance

relationships across layers throughout training:

$$\mathrm{Cov}(\widehat{Z}^{W_0^l \delta x_s^{l-1}}(\xi), \widehat{Z}^{W_0^l \delta x_t^{l-1}}(\varsigma)) = \mathbb{E}[Z^{\delta x_s^{l-1}}(\xi) Z^{\delta x_t^{l-1}}(\varsigma)],$$

$$\mathrm{Cov}(\widehat{Z}^{W_0^{l\top} dh_s^l}(\xi), \widehat{Z}^{W_0^{l\top} dh_t^l}(\varsigma)) = \mathbb{E}[Z^{dh_s^l}(\xi) Z^{dh_t^l}(\varsigma)].$$

These covariance relationships reveal that feature correlations between adjacent layers follow consistent patterns, even as individual features evolve. They link the feature spaces of adjacent layers through their second-order statistics, providing a structural bridge that persists throughout the training process. While previous work has investigated covariance structures in neural networks (Pandey et al., 2022; Guth et al., 2024)—these studies do not explicitly examine the interaction of the covariance matrix across layers during training dynamics. Our analysis reveals how these structural invariants enable both feature learning and global convergence under $\mu$P.

### 5.3. From Covariance Structure to Non-degeneracy

The preservation of covariance relationships across layers ensures the non-degeneracy of the induced Gaussian processes throughout training. In the proof of Theorem 4.5, we consider any linear combination of the Gaussian processes:

$$\sum_{i\in[m],s\in[t]} \lambda_{i,s}\widehat{Z}^{W_0^l \delta x_s^{l-1}}(\xi_i), \quad \sum_{i\in[m],s\in[t]} \lambda_{i,s}\widehat{Z}^{W_0^{l\top} dh_s^l}(\xi_i).$$

Through our covariance preservation property, we show that if these linear combinations degenerate (i.e., equal to zero almost surely), then the corresponding linear combinations of original features and gradients must also degenerate:

$$\sum_{i\in[m],s\in[t]} \lambda_{i,s}Z^{\delta x_s^{l-1}}(\xi_i) \overset{a.s.}{=} 0,$$

$$\sum_{i\in[m],s\in[t]} \lambda_{i,s}Z^{dh_s^l}(\xi_i) \overset{a.s.}{=} 0.$$

This connection through linear combinations allows us to transfer the non-degeneracy property from feature space to the induced Gaussian processes across layers, establishing that both forward and backward processes remain non-degenerate throughout training. This result reveals a fundamental connection between covariance structure and feature richness: the preservation of covariance relationships ensures that linear independence propagates through layers.

**This directly contrasts with other parametrizations.** In the NTK parametrization, since features stay close to initialization with $Z^{z_t}(\xi) = Z^{z_0}(\xi)$, the process necessarily becomes degenerate as it fails to capture new information during training. Our analysis can demonstrate that $\mu$P uniquely maintains the non-degeneracy of features across both space and time dimensions, enabling the network to learning rich and meaningful features throughout training.

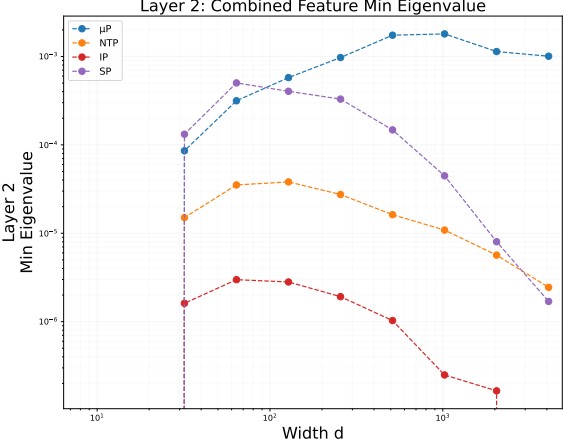

*Figure 2.* Layer 2: Minimum eigenvalue analysis of pre-activation features by concatenating initial and final states across different parametrizations in 3 hidden-layer MLPs trained on CIFAR-10. The $\mu$P parametrization maintains higher eigenvalues as width increases, demonstrating better preservation of feature richness. In contrast, other parametrizations (NTP, IP, SP) show substantial decay in eigenvalues with increasing width, indicating feature degeneration. This empirically validates our theoretical analysis that $\mu$P uniquely preserves non-degeneracy across both spatial and temporal dimensions, while NTP and SP fail to capture new information during training.

To empirically validate this theoretical finding, we analyze the minimum eigenvalue of the feature matrix constructed from joint space-time features at Layer 2, complementing our analysis of feature diversity in Figure 1. Under the same experimental setup with 3 hidden-layer MLPs trained on CIFAR-10, Figure 2 constructs joint space-time representations: for each input $\xi_i$, we concatenate initial features $h^0(\xi_i)$ and final features $h^T(\xi_i)$ to form the combined matrix $[h_1^0, h_1^T, h_2^0, h_2^T, \ldots, h_N^0, h_N^T]$, then compute the minimum eigenvalue of its Gram matrix. The results show that $\mu$P maintains higher eigenvalues across different network widths compared to other parametrizations.

This aligns with our theoretical prediction and further strengthens the findings in Figure 1 where we observed $\mu$P's unique ability to achieve both feature learning and feature richness. Additional comprehensive eigenvalue spectrum analysis across all layers and both pre- and post-activation features is provided in Appendix A.1, A.2, and A.3.

### 5.4. Evolution Framework

To rigorously track how these structural properties evolve throughout training, we need to carefully handle the natural flow of information in neural networks: forward propagation followed by backward propagation. In each iteration, the network first computes forward features through all layers, then calculates gradients backwards for parameter updates.

This computational pattern naturally leads to a two-level filtration framework. We introduce a sequence of $\sigma$-algebras to track the evolution of random variables during training. Let $\mathcal{F}_0$ denote the initial condition:

$$\mathcal{F}_0 = \sigma\big(\{Z^{h_0^l(\xi_i)}, Z^{x_0^l(\xi_i)}\}_{i \in [m], l \in [L]}, Z^{\widehat{W}_0^{L+1}}\big)$$

Then we define $\mathcal{F}_t$ to track all completed iterations up to time $t$, and an extended filtration $\mathcal{G}_t$ to capture the forward pass of the $(t+1)$-th iteration:

$$\mathcal{F}_t = \sigma\big(\mathcal{F}_0, \{\widehat{Z}^{W_0^l \delta x_s^{l-1}(\xi_i)}\}_{i \in [m], s \in [t], 2 \leq l \leq L},$$
$$\{\widehat{Z}^{W_0^{l\top} dh_s^l(\xi_i)}\}_{i \in [m], s \in [t], 2 \leq l \leq L}\big) \qquad (5.5)$$

$$\mathcal{G}_t = \sigma\big(\mathcal{F}_t, \{\widehat{Z}^{W_0^l \delta x_{t+1}^{l-1}(\xi_i)}\}_{i \in [m], 2 \leq l \leq L}\big) \qquad (5.6)$$

This filtration structure allows us to precisely track how information flows through the network: $\mathcal{F}_t$ contains all information up to time $t$, while $\mathcal{G}_t$ extends this to include the forward pass information at time $t+1$ before its backward pass begins. This framework enables us to:

1. **Inductive Proof Structure:** The filtration framework enables a structured inductive proof that follows the natural flow of computation in neural networks. We establish non-degeneracy in four steps, motivated by how information propagates through the network:

   - Step 1: Features in first hidden layer $\widehat{Z}^{W_0^2 \delta x_s^1(\xi_i)}$. This forms the base case as it only depends on the input,
   - Step 2: Features in remaining layers $\widehat{Z}^{W_0^l \delta x_s^{l-1}(\xi_i)}$. Using the non-degeneracy of previous layers,
   - Step 3: Gradients in last layer $\widehat{Z}^{W_0^{L\top} dh_s^L(\xi_i)}$. Built upon the established feature properties,
   - Step 4: Gradients in remaining layers $\widehat{Z}^{W_0^{l\top} dh_s^l(\xi_i)}$. Completing the backward pass analysis.

   Each step leverages the non-degeneracy established in previous steps, creating a chain of dependency that mirrors the network's computation graph.

2. **Conditional Analysis:** The filtration enables precise decomposition of feature and gradient updates into new and historical information:

$$Z^{h_s^l(\xi_i)} = \underbrace{\widehat{Z}^{W_0^l \delta x_s^{l-1}(\xi_i)}}_{\text{new randomness}} + \underbrace{\Delta_s(\xi_i)}_{\text{history}}$$

$$Z^{dx_s^l(\xi_i)} = \underbrace{\widehat{Z}^{W_0^{l+1\top} dh_s^{l+1}(\xi_i)}}_{\text{new randomness}} + \underbrace{G_s(\xi_i)}_{\text{history}}$$

   where $\Delta_s(\xi_i) \in \mathcal{F}_{s-1}$ captures the accumulated feature history, and $G_s(\xi_i)$ represents previous gradient information. This decomposition is crucial for our inductive proof: by focusing on the **new randomness** in each step, we can show that non-degeneracy is preserved when conditioned on all historical information.

3. **Non-degeneracy Preservation:** By leveraging GOOD activation functions as introduced in Assumption 4.3 (e.g., Sigmoid, Tanh, SiLU) and the covariance structure, we show that non-degeneracy propagates forward in time. Specifically, if at time $t$ we have:

$$\sum_{i \in [m], s \in [t]} \lambda_{i,s} \widehat{Z}^{W_0^l \delta x_s^{l-1}(\xi_i)} \overset{a.s.}{\neq} 0$$

$$\sum_{i \in [m], s \in [t]} \lambda_{i,s} \widehat{Z}^{W_0^{l\top} dh_s^l(\xi_i)} \overset{a.s.}{\neq} 0$$

Then, we show that these sums remain nonzero at time $t+1$ based on two key properties: (1) the non-degeneracy of Gaussian processes is preserved when they share the same covariance structure, and (2) GOOD activation functions, such as Sigmoid and SiLU, exhibit a crucial "non-collapsing" property—mapping distinct inputs to distinct outputs unless all combining coefficients are zero. Together, these properties ensure that features can evolve significantly during training while preserving their diversity, a fundamental distinction from the NTK parametrization, where features remain near their initialization.

Now, we revisit the key technical challenges introduced at the beginning of this section and demonstrate how our framework addresses them.

**Feature Evolution vs. Structural Stability:** Unlike NTK, where features remain close to their initialization, $\mu$P enables substantial feature evolution. Our framework ensures that new randomness, represented by Gaussian features such as $\widehat{Z}^{W_0^l \delta x_{t+1}^{l-1}(\xi_i)}$, enters the system with a well-defined structure that preserves non-degeneracy. This structured evolution prevents feature collapse while allowing representations to adapt dynamically, ensuring both expressivity and stability throughout training.

**Cross-Layer Coupling:** The interplay between layers introduces dependencies that can destabilize training. By leveraging a two-level filtration structure $\mathcal{F}_t$ and $\mathcal{G}_t$, our framework tracks both forward propagation $Z^{h_s^l}$ and backward propagation $Z^{dx_s^l}$, ensuring that updates in one layer do not collapse the feature space of others. This structure maintains well-defined covariance relationships across layers, allowing $\mu$P to support both deep feature learning and global convergence, distinguishing it from NTK and standard parametrizations.

## 6. Conclusion and Future Work

In this work, we establish a fundamental theoretical result: deep neural networks under $\mu$P parametrization can simultaneously achieve meaningful feature learning while preserving feature non-degeneracy. Through a rigorous analysis of

Gaussian processes and their covariance structures, we show that features not only remain linearly independent throughout training but also undergo substantial evolution from their initialization. This provides insight into a fundamental question in deep learning theory: how neural networks can simultaneously learn expressive representations and achieve global convergence.

Our analysis establishes fundamental connections between covariance preservation and feature richness. By preventing feature degeneracy, our framework provides a rigorous foundation for understanding how overparameterized networks learn expressive representations. Moreover, our results highlight the crucial role of parametrization in enabling both stable training and meaningful feature evolution. These insights into how $\mu$P enables both feature learning and global convergence suggest promising directions for bridging the gap between theory and practical deep learning success.

Several promising directions for future work emerge from our analysis. First, extending our theoretical framework to transformer architectures, particularly the attention mechanism, would be valuable for understanding feature learning in modern language models. Second, our analysis of structural invariants could provide new perspectives on convergence rates beyond just global convergence, potentially informing optimization strategies in deep learning. Third, studying how our insights on feature non-degeneracy influence generalization bounds may yield deeper theoretical foundations for understanding the generalization properties of deep neural networks. Finally, exploring how $\mu$P interacts with more complex training paradigms, such as fine-tuning and self-supervised learning, could further enhance our understanding of deep network training dynamics in practical settings. Additionally, extending our convergence analysis to continuous-depth architectures such as neural ODEs and residual networks (Yang et al., 2023b; Marion et al., 2023; Bordelon et al., 2023; Gao et al.) represents an interesting direction for future research.

## Acknowledgements

We thank the anonymous reviewers and area chair for their helpful comments. ZC is supported by the UCLA Dissertation Year Fellowship. QZ and QG are supported in part by the National Science Foundation CAREER Award 1906169 and IIS-2008981, as well as the Sloan Research Fellowship. The views and conclusions contained in this paper are those of the authors and should not be interpreted as representing any funding agencies.

## Impact Statement

This work develops theoretical foundations for understanding feature learning in deep neural networks, focusing on how different parametrization schemes affect the learning dynamics. While primarily theoretical in nature, our findings may inform the design of more efficient training algorithms and architectures. As our contribution focuses on mathematical foundations, its societal impact aligns with the well-established consequences of advancing machine learning research.

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

# A. Experimental Details

**Experimental details for Figures 1 and 2.** We conduct experiments using MLPs with three hidden layers and input dimension $n_0 = 3072$ (flattened CIFAR-10 images), three hidden layers of equal width $n_1 = n_2 = n_3 = n$ varying from $8, 16, 32, 64, 128, 256, 512, 1024, 2048, 4096$, and output dimension 1. In the experiment, we only use 10 samples randomly selected from the airplane and automobile classes in CIFAR-10. All networks use SiLU activation functions and are trained on a binary classification task with $\pm 1$ targets for 1000 steps. We use a global learning rate $\eta = 0.1$ across all parametrization schemes with 10 runs for each width setting. The learning rate $\eta = 0.1$ is chosen to ensure stable training across parametrizations.

We implement the following parametrization schemes:

**Standard Parametrization (SP):**

$$\sigma_\ell = \sqrt{\frac{2}{n_{\ell-1}}}; \quad \eta_\ell = \eta \cdot \frac{1}{n}$$

at all layers, with $\eta = 0.1$.

**Neural Tangent Parametrization (NTP):**

$$\sigma_\ell = \sqrt{\frac{2}{n_{\ell-1}}}; \quad \eta_\ell = \eta \cdot \frac{1}{n_{\ell-1}}$$

at all layers, with $\eta = 0.1$.

**Integrable Parametrization (IP):** For initialization variances:

$$\sigma_1 = \sqrt{\frac{2}{d}}; \quad \sigma_\ell = \frac{\sqrt{2}}{n}, \quad \ell \geq 2.$$

For learning rates:

$$\eta_1 = \eta \cdot \frac{n}{d}; \quad \eta_2 = \eta_3 = \eta; \quad \eta_4 = \eta \cdot \frac{1}{n}.$$

**Maximal Update Parametrization ($\mu$P):** For initialization variances:

$$\sigma_1 = \sqrt{\frac{2}{d}}; \quad \sigma_2 = \sigma_3 = \sqrt{\frac{2}{n}}; \quad \sigma_4 = \frac{\sqrt{2}}{n}.$$

For learning rates:

$$\eta_1 = \eta \cdot \frac{n}{d}; \quad \eta_2 = \eta_3 = \eta; \quad \eta_4 = \eta \cdot \frac{1}{n}$$

Networks are trained with batch size 10 for 1000 steps, sufficient for all widths to achieve stable feature representations with training loss smaller than 0.05. For each width configuration, we conduct 10 independent trials with different random seeds $(42 \sim 53)$ and report the mean values in Figures 1 and 2. To quantify feature properties, we measure two metrics:

- Feature change: $\|h(x) - h^0(x)\|_2 / \|h^0(x)\|_2$, where $h^0$ represents features at initialization

- Feature diversity: minimum eigenvalue of the Gram matrix $K_{ij} = \langle h(x_i), h(x_j) \rangle$ computed over batch samples

These measurements allow us to track both the evolution of features from their initialization state and the maintenance of feature richness throughout training.

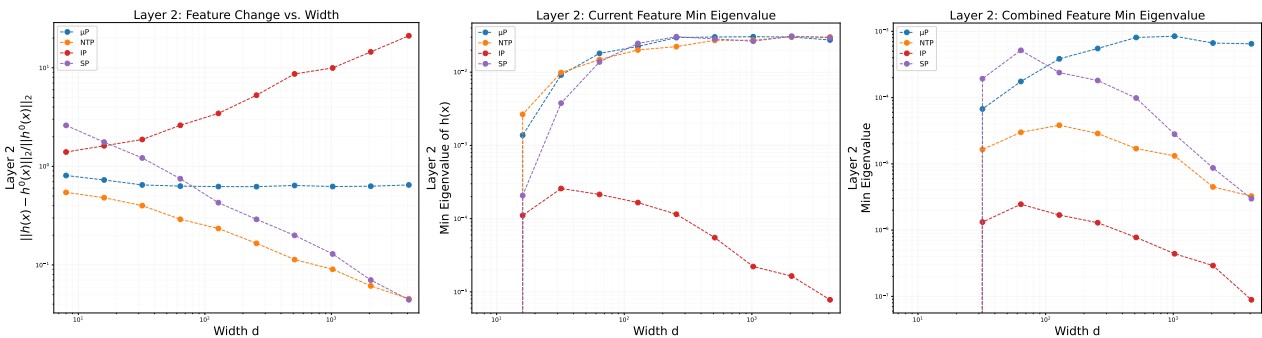

*Figure 3.* Feature learning behavior for Tanh activation. **Left:** Feature change ($\|h(x) - h^0(x)\|_2 / \|h^0(x)\|_2$). **Middle:** Feature diversity (minimum eigenvalue of Gram matrix). **Right:** Minimum eigenvalue analysis by concatenating initial and final features.

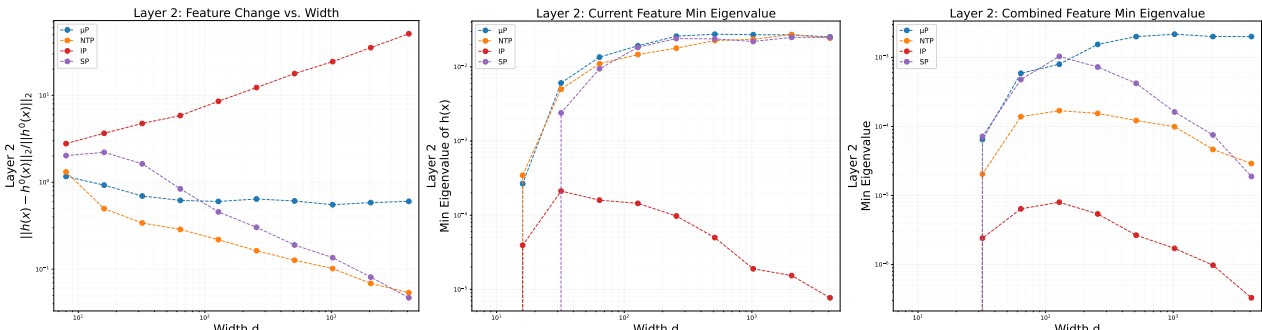

*Figure 4.* Feature learning behavior for ReLU activation. **Left:** Feature change ($\|h(x) - h^0(x)\|_2 / \|h^0(x)\|_2$). **Middle:** Feature diversity (minimum eigenvalue of Gram matrix). **Right:** Minimum eigenvalue analysis by concatenating initial and final features.

### A.1. Additional Results on Activation Functions

To further examine the impact of activation functions, we conduct experiments with Tanh and ReLU under the same settings as described in Section A. Below, we present the feature evolution and diversity results for these activations.

Our theoretical analysis fully explains the feature learning behavior of Tanh networks, as confirmed by our experimental results in Figure 3. Tanh enables meaningful feature learning while leading to a gradual decrease in feature diversity as width goes up. This is consistent with our theoretical predictions, which account for the smooth and bounded nature of the Tanh activation.

While our theoretical analysis does not directly apply to ReLU due to its non-smooth nature, our experimental results in Figure 4 indicate that ReLU-trained networks still exhibit feature learning and maintain meaningful representations under Maximal Update Parametrization ($\mu$P). One possible explanation is that, although ReLU lacks explicit smoothness assumptions required in our analysis, its piecewise linear structure still allows for non-trivial feature evolution in practice. Moreover, $\mu$P ensures that weight updates are appropriately scaled across layers, preventing degenerate training dynamics that could otherwise hinder learning in deep networks. Understanding the precise mechanisms behind ReLU's feature learning in the infinite-width setting remains an important direction for future theoretical work.

## A.2. Additional Results on Eigenvalue Spectrum Analysis

### A.2.1. POST-ACTIVATION EIGENVALUE SPECTRUM

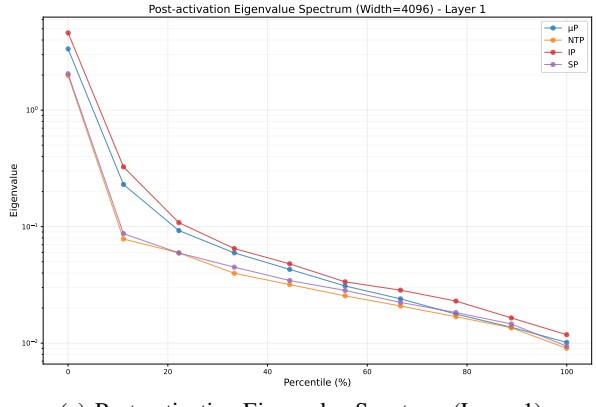

(a) Post-activation Eigenvalue Spectrum (Layer 1)

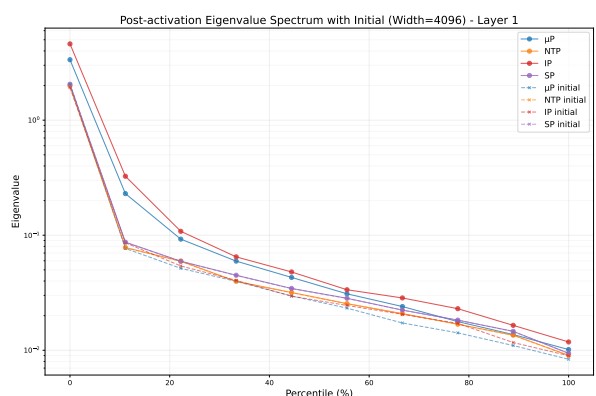

(b) Post-activation Eigenvalue Spectrum with Initial (Layer 1)

*Figure 5.* Layer 1 Post-activation Eigenvalue Spectrum Analysis. The x-axis shows percentiles of the sorted eigenvalues of the feature Gram matrix, where 0% represents the largest eigenvalue and 100% the smallest. The y-axis (log scale) shows the magnitude of these eigenvalues. $\mu$P and IP maintain higher eigenvalues throughout the spectrum compared to NTP and SP, with the right plot including initialization values (dashed lines) for comparison.

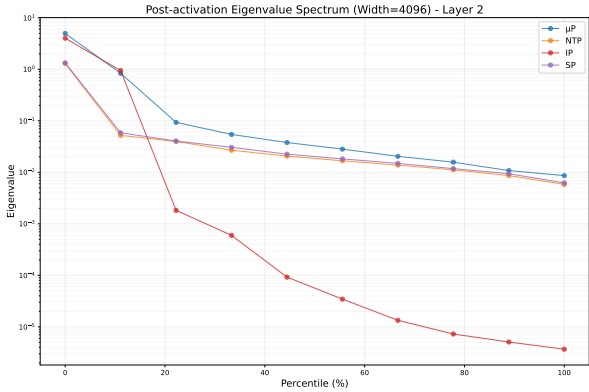

(a) Post-activation Eigenvalue Spectrum (Layer 2)

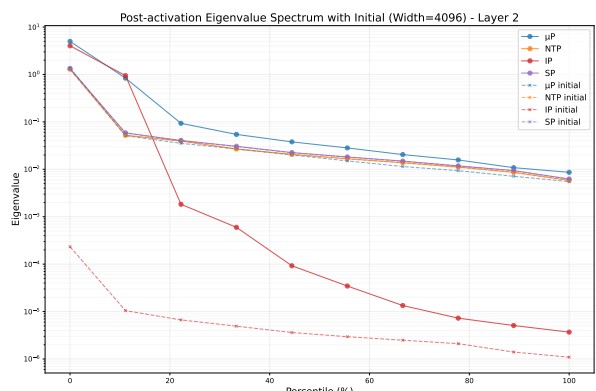

(b) Post-activation Eigenvalue Spectrum with Initial (Layer 2)

*Figure 6.* Layer 2 Post-activation Eigenvalue Spectrum Analysis. The plots display eigenvalues of the feature Gram matrix ranked by percentile (0% = largest, 100% = smallest). IP shows dramatic eigenvalue collapse at higher percentiles (smaller eigenvalues, right side of plot), while $\mu$P maintains substantially higher eigenvalues across all percentiles, indicating preserved feature diversity even among the least significant dimensions.

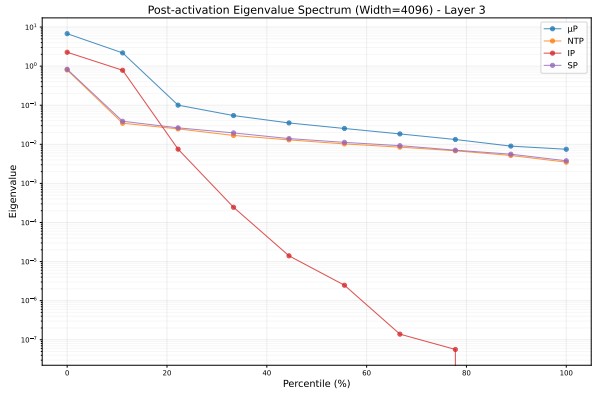 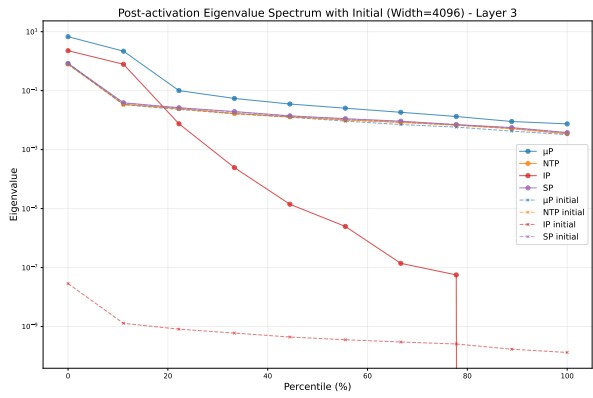

(a) Post-activation Eigenvalue Spectrum (Layer 3)  (b) Post-activation Eigenvalue Spectrum with Initial (Layer 3)

*Figure 7.* Layer 3 Post-activation Eigenvalue Spectrum Analysis. The x-axis shows percentiles of ranked eigenvalues, with 100% representing the smallest eigenvalue. In this deepest layer, IP exhibits catastrophic collapse, with eigenvalues at higher percentiles (60-100%) plummeting to $10^{-7}$. In contrast, $\mu$P maintains eigenvalues orders of magnitude larger across all percentiles, demonstrating superior feature independence throughout the entire spectrum.

A.2.2. PRE-ACTIVATION EIGENVALUE SPECTRUM

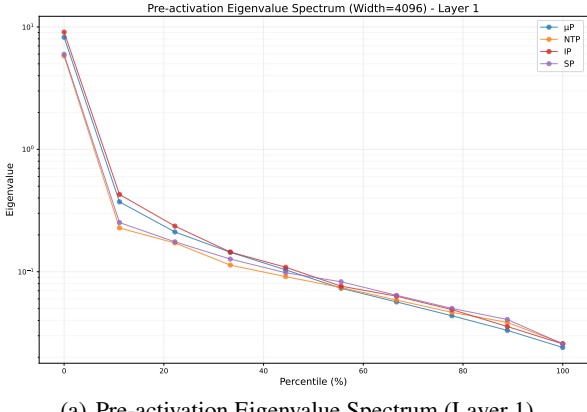 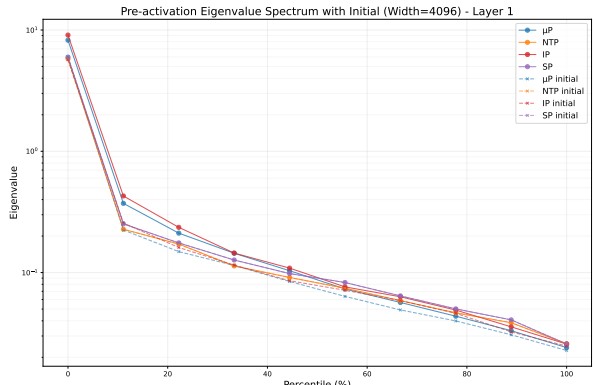

(a) Pre-activation Eigenvalue Spectrum (Layer 1)  (b) Pre-activation Eigenvalue Spectrum with Initial (Layer 1)

*Figure 8.* Layer 1 Pre-activation Eigenvalue Spectrum Analysis. The eigenvalues of the feature Gram matrix are plotted from largest (0% percentile) to smallest (100% percentile). Higher eigenvalues, particularly at higher percentiles (right side), indicate greater feature diversity and less redundancy. $\mu$P maintains strong feature independence throughout training compared to other parameterizations.

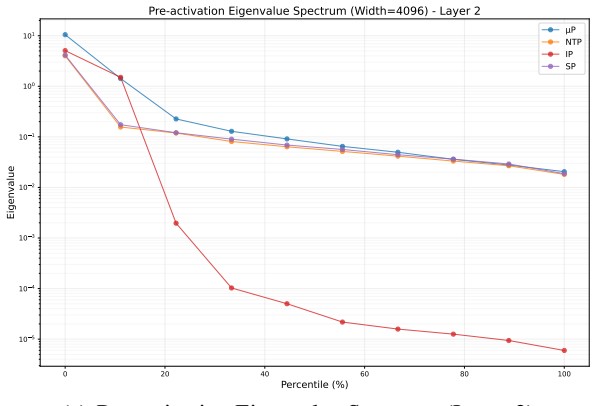

(a) Pre-activation Eigenvalue Spectrum (Layer 2)

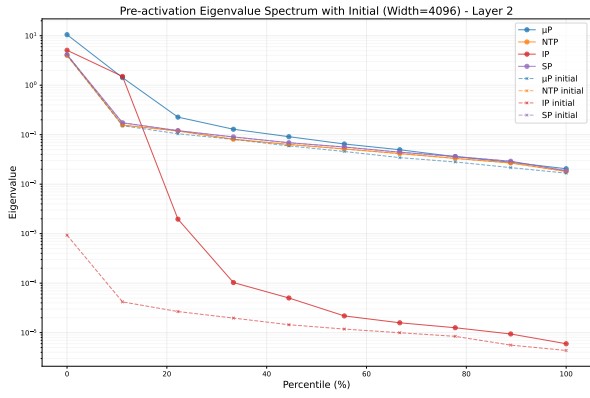

(b) Pre-activation Eigenvalue Spectrum with Initial (Layer 2)

*Figure 9.* Layer 2 Pre-activation Eigenvalue Spectrum Analysis. Eigenvalues of the feature Gram matrix are ranked by percentile on the x-axis. The smallest eigenvalues (highest percentiles, right side) are particularly important as they indicate linear independence among features. $\mu$P maintains significantly higher eigenvalues at all percentiles compared to NTP and SP, while IP experiences severe feature collapse above the 20th percentile.

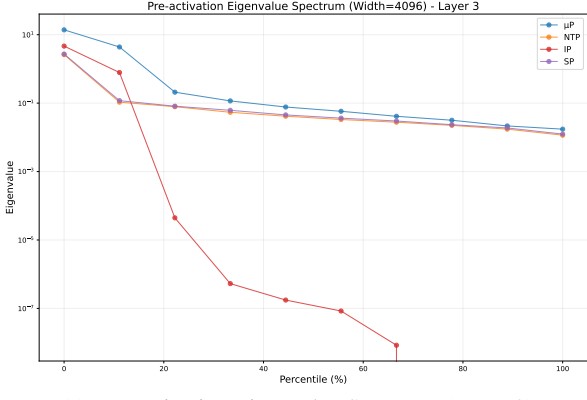

(a) Pre-activation Eigenvalue Spectrum (Layer 3)

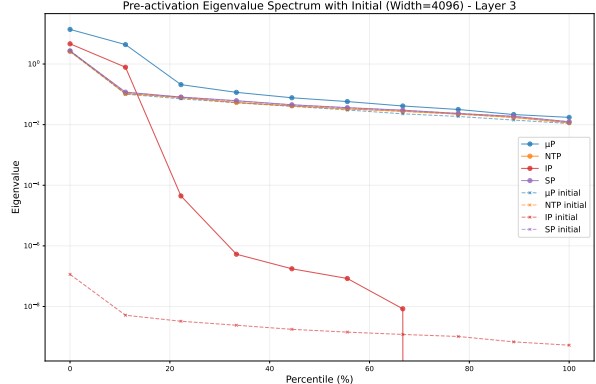

(b) Pre-activation Eigenvalue Spectrum with Initial (Layer 3)

*Figure 10.* Layer 3 Pre-activation Eigenvalue Spectrum Analysis. Eigenvalues are arranged by percentile rank, from largest (0%) to smallest (100%). At this deepest layer, the percentile-based analysis reveals the most dramatic differentiation between parameterizations. $\mu$P maintains substantial eigenvalues even at the highest percentiles (80-100%), while IP experiences catastrophic collapse beyond the 20th percentile. The right plot shows initialization values (dashed lines) for comparison with final trained features.

## A.3. Feature Learning and Independence Metrics

### A.3.1. POST-ACTIVATION FEATURES

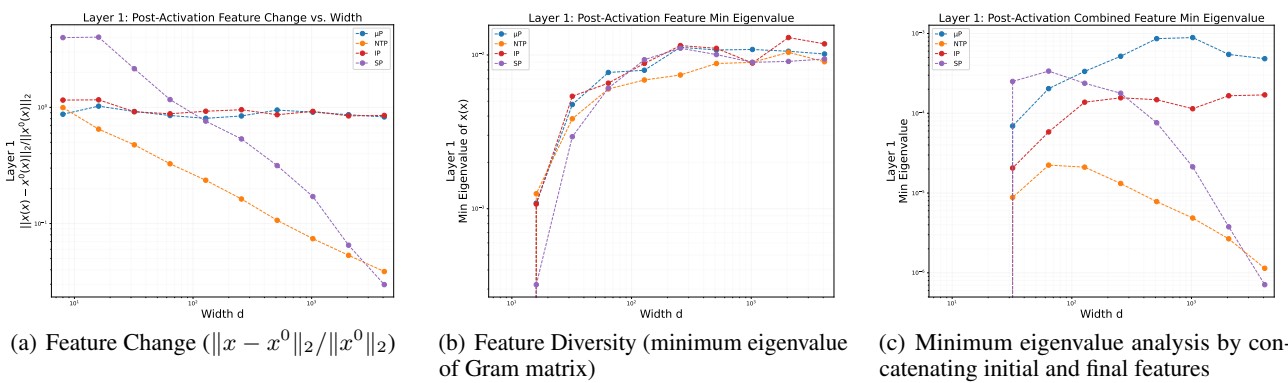

(a) Feature Change ($\|x - x^0\|_2 / \|x^0\|_2$)

(b) Feature Diversity (minimum eigenvalue of Gram matrix)

(c) Minimum eigenvalue analysis by concatenating initial and final features

*Figure 11.* Post-Activation feature learning behavior for SILU activation in Layer 1. **Left:** Feature change ($\|x - x^0\|_2 / \|x^0\|_2$). **Middle:** Feature diversity (minimum eigenvalue of Gram matrix). **Right:** Minimum eigenvalue analysis by concatenating initial and final features.

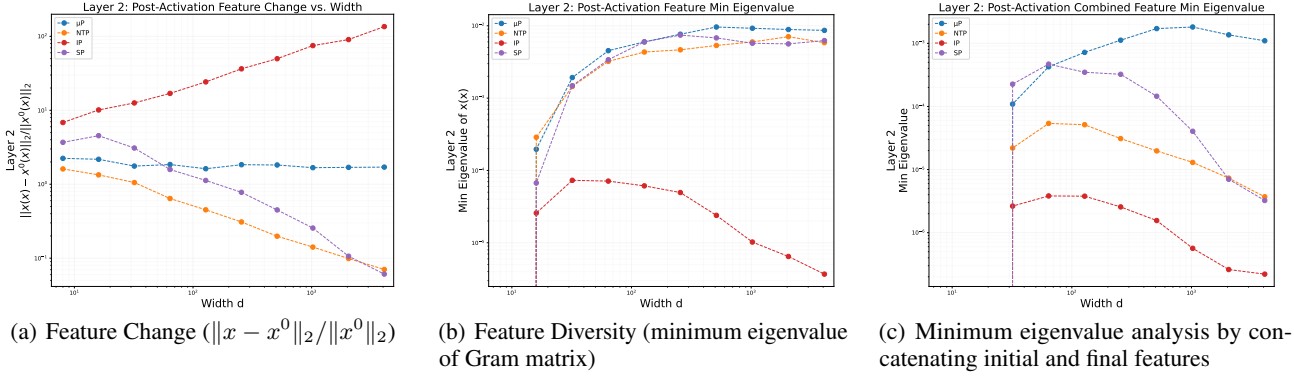

(a) Feature Change ($\|x - x^0\|_2 / \|x^0\|_2$)

(b) Feature Diversity (minimum eigenvalue of Gram matrix)

(c) Minimum eigenvalue analysis by concatenating initial and final features

*Figure 12.* Post-Activation feature learning behavior for SILU activation in Layer 2. **Left:** Feature change ($\|x - x^0\|_2 / \|x^0\|_2$). **Middle:** Feature diversity (minimum eigenvalue of Gram matrix). **Right:** Minimum eigenvalue analysis by concatenating initial and final features.

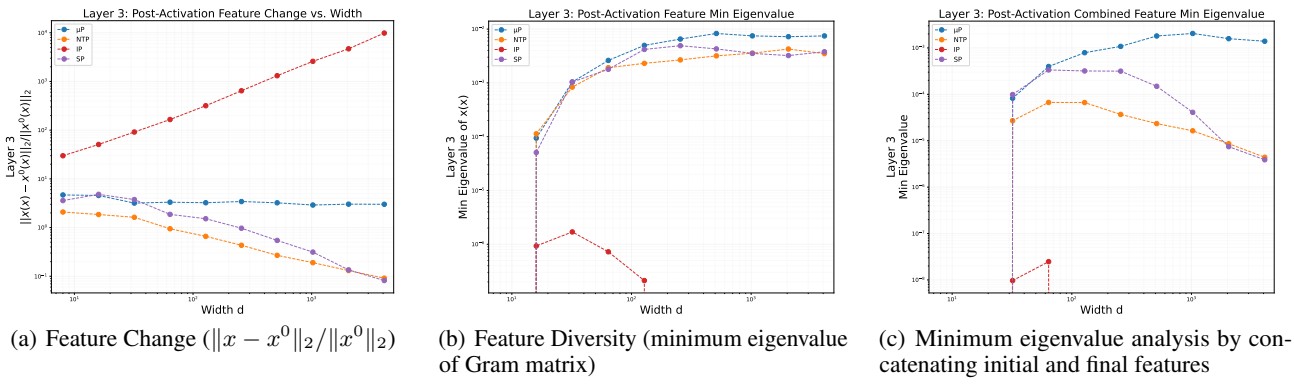

(a) Feature Change ($\|x - x^0\|_2 / \|x^0\|_2$)

(b) Feature Diversity (minimum eigenvalue of Gram matrix)

(c) Minimum eigenvalue analysis by concatenating initial and final features

*Figure 13.* Post-Activation feature learning behavior for SILU activation in Layer 3. **Left:** Feature change ($\|x - x^0\|_2 / \|x^0\|_2$). **Middle:** Feature diversity (minimum eigenvalue of Gram matrix). **Right:** Minimum eigenvalue analysis by concatenating initial and final features.

### A.3.2. PRE-ACTIVATION FEATURES

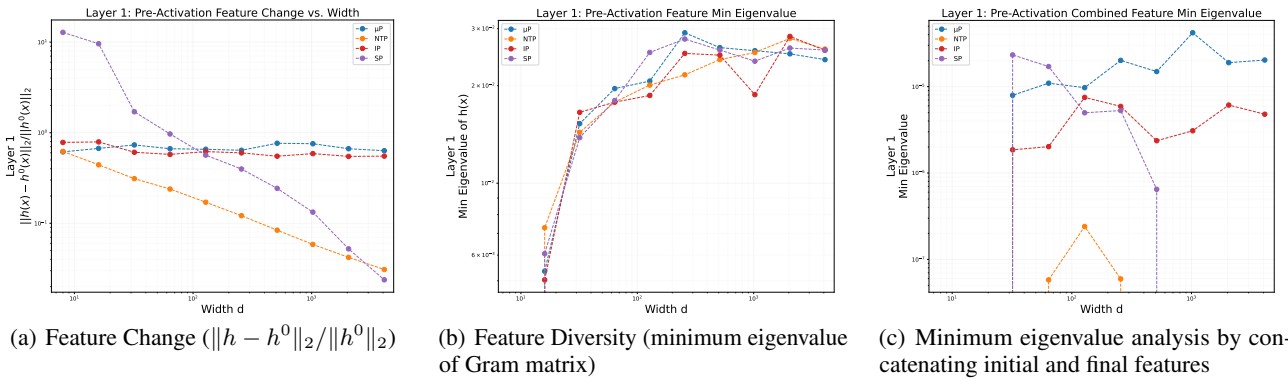

(a) Feature Change ($\|h - h^0\|_2/\|h^0\|_2$)

(b) Feature Diversity (minimum eigenvalue of Gram matrix)

(c) Minimum eigenvalue analysis by concatenating initial and final features

*Figure 14.* Pre-Activation feature learning behavior for SILU activation in Layer 1. **Left:** Feature change ($\|h - h^0\|_2/\|h^0\|_2$). **Middle:** Feature diversity (minimum eigenvalue of Gram matrix). **Right:** Minimum eigenvalue analysis by concatenating initial and final features.

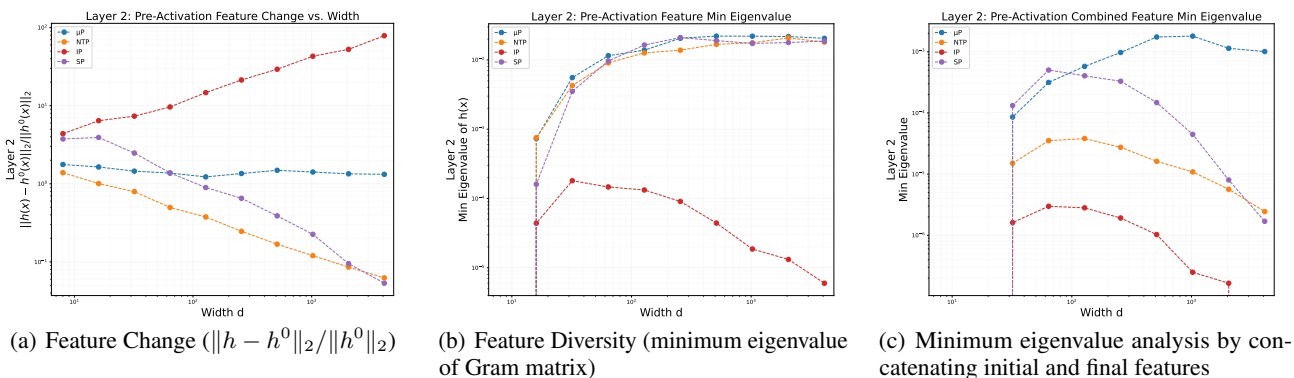

(a) Feature Change ($\|h - h^0\|_2/\|h^0\|_2$)

(b) Feature Diversity (minimum eigenvalue of Gram matrix)

(c) Minimum eigenvalue analysis by concatenating initial and final features

*Figure 15.* Pre-Activation feature learning behavior for SILU activation in Layer 2. **Left:** Feature change ($\|h - h^0\|_2/\|h^0\|_2$). **Middle:** Feature diversity (minimum eigenvalue of Gram matrix). **Right:** Minimum eigenvalue analysis by concatenating initial and final features.

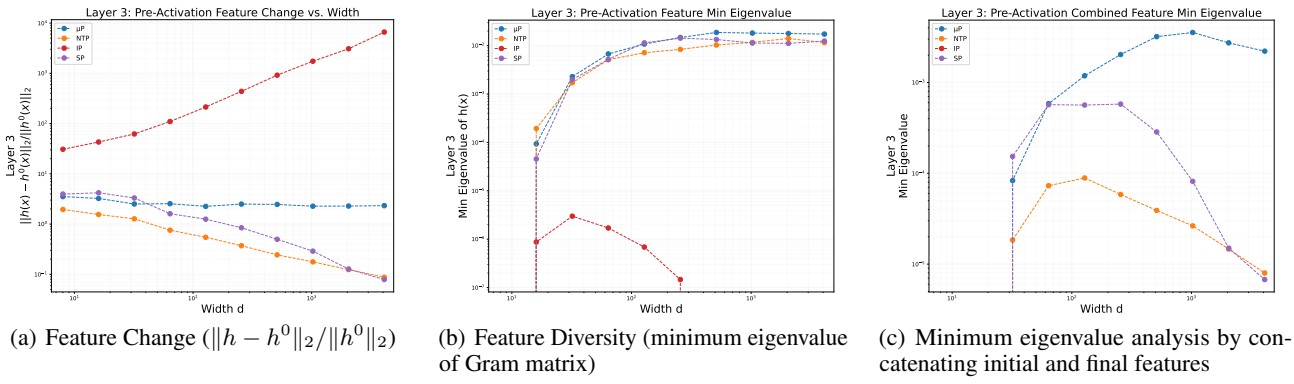

(a) Feature Change ($\|h - h^0\|_2/\|h^0\|_2$)

(b) Feature Diversity (minimum eigenvalue of Gram matrix)

(c) Minimum eigenvalue analysis by concatenating initial and final features

*Figure 16.* Pre-Activation feature learning behavior for SILU activation in Layer 3. **Left:** Feature change ($\|h - h^0\|_2/\|h^0\|_2$). **Middle:** Feature diversity (minimum eigenvalue of Gram matrix). **Right:** Minimum eigenvalue analysis by concatenating initial and final features.

# B. More Details for $\mu$P Parametrization

Formally, the MLP definition (Yang & Hu, 2020, Table 1) in this section is

$$h^1 = W\xi \in \mathbb{R}^n, x^l = \phi(h^l) \in \mathbb{R}^n, h^{l+1} = W^{l+1}x^l \in \mathbb{R}^n, f(\xi) = W^{L+1}x^L, \tag{B.1}$$

where $L > 1$ is any positive integer and $l \in \{1, \ldots, L-1\}$. Then the $\mu$P for this $L$-hidden-layer MLP is defined as follows (Yang & Hu, 2020).

1. Initial weight matrices in the middle layer: $W_0^2, \ldots, W_0^L$, with each coordinates $(W_0^l)_{\alpha\beta} \sim \mathcal{N}(0, 1/n)$.

2. Initial weight matrix in the input and output layers: input layer matrix $W_0^1 \in \mathbb{R}^{n \times d}$ and output layer matrix $\widehat{W}_0^{L+1} := W_0^{L+1}n \in \mathbb{R}^{1 \times n}$, with each coordinate $(W_0^1)_{\alpha\beta}, (\widehat{W}_0^{L+1})_{\alpha\beta} \sim \mathcal{N}(0, 1)$.

3. Initial model outputs: we define the scalars $f_0(\xi) := W_0^{L+1}x_0^L(\xi)$ for any input $\xi$.

Assuming the same Assumption 4.3 for $\phi$, we can characterize the $Z$ variables in the *infinite-width* training dynamics of SGD for this $L$-hidden-layer MLP similarly as follows (Yang & Hu, 2020).

1. For $z \in \{x^l, h^l\}_l$, we have

$$Z^{z_t(\xi)} = Z^{z_0(\xi)} + Z^{\delta z_1(\xi)} + \cdots Z^{\delta z_t(\xi)} \tag{B.2}$$

2. For $l \in [L], x = x^l, h = h^l$, we have

$$Z^{\delta x_t(\xi)} = \phi(Z^{h_t(\xi)}) - \phi(Z^{h_{t-1}(\xi)}). \tag{B.3}$$

3. For $h = h^1$, we have

$$Z^{\delta h_t(\xi)} = -\sum_{i \in [m]} \eta \mathring{\chi}_{t-1,i} \xi_i^\top \xi Z^{dh_{t-1}(\xi_i)}.$$

4. For $2 \leq l \leq L, h = h^l, x = x^{l-1}, W = W^l$, we have

$$Z^{\delta h_t(\xi)} = \widehat{Z}^{W_0 \delta x_t(\xi)} + \dot{Z}^{W_0 \delta x_t(\xi)} - \eta \sum_{s=0}^{t-1} \sum_{i \in [m]} \mathring{\chi}_s Z^{dh_s(\xi_i)} \mathbb{E} Z^{x_s(\xi_i)} Z^{x_t(\xi)}$$

where

$$\dot{Z}^{W_0 \delta x_t(\xi)} = \sum_{i \in [m]} \sum_{s=0}^{t-1} Z^{dh_s(\xi_i)} \mathbb{E} \frac{\partial Z^{\delta x_t(\xi)}}{\partial \widehat{Z}^{W_0^\top dh_s(\xi_i)}}.$$

5. For last layer weight

$$Z^{\widehat{W}_t^{L+1}} = Z^{\widehat{W}_0^{L+1}} - \eta \sum_{s=0}^{t-1} \sum_{i \in [m]} \mathring{\chi}_{s,i} Z^{x_s^L(\xi_i)} \tag{B.4}$$

6. The output deltas have limits

$$\delta \mathring{f}_t(\xi) = \mathbb{E} Z^{\delta W_t^{L+1}} Z^{x_t^L(\xi)} + \mathbb{E} Z^{\widehat{W}_{t-1}^{L+1}} Z^{\delta x_t^L(\xi)} \tag{B.5}$$

and

$$\mathring{f}_t(\xi) = \delta \mathring{f}_1(\xi) + \cdots + \delta \mathring{f}_t(\xi).$$

7. For gradients:

$$Z^{dx_t^L(\xi)} = Z^{\widehat{W}_t^{L+1}} \tag{B.6}$$

$$Z^{dh_t^l(\xi)} = Z^{dx_t^l(\xi)} \phi'(Z^{h_t^l(\xi)}) \tag{B.7}$$

$$Z^{dx_t^{l-1}(\xi)} = \widehat{Z}^{W_0^{l\top} dh_t^l(\xi)} + \dot{Z}^{W_0^{l\top} dh_t^l(\xi)} - \eta \sum_{s=0}^{t-1} \sum_{i \in [m]} \mathring{\chi}_{s,i} Z^{x_s^{l-1}(\xi_i)} \mathbb{E} Z^{dh_s^l(\xi_i)} Z^{dh_t^l(\xi)} \tag{B.8}$$

where

$$\dot{Z}^{W_0^{l\top} dh_t^l(\xi)} = \sum_{i \in [m]} \sum_{s=0}^{t-1} Z^{x_s^{l-1}(\xi_i)} \mathbb{E} \frac{\partial Z^{dh_t^l(\xi)}}{\partial \widehat{Z}^{W_0^l x_s^{l-1}(\xi_i)}}.$$

8. Loss derivative:

$$\mathring{\chi}_{t,i} = \mathcal{L}'(\mathring{f}_t, y_i, t, i) = \mathcal{L}'(\mathring{f}_t, y_i) \mathbb{1}\{i \in \mathcal{B}_t\}.$$

**Intuition behind the entanglement term for the two-hidden-layer case** The inclusion of $W^\top$ in the backward pass largely increases the system complexity by introducing multiplications between $W$ and certain nonlinear transformations of $W^\top$ in the forward pass, which necessitates involved definitions of $\dot{Z}^{Wx_t(\xi)}$ and $\dot{Z}^{W^\top d\bar{h}_t(\xi)}$. Since they are all "conditioned out" in our analysis, we only showcase the definition of $\dot{Z}^{Wx_t(\xi)} = \sum_{j=1}^m \sum_{r=0}^{t-1} \theta_{r,j} Z^{d\bar{h}_r(\xi_j)}$ to give a *sense of entanglement* between $W$ and $W^\top$, where $\theta_r$ is calculated like so: $Z^{x_t(\xi)}$ by definition is constructed as

$$Z^{x_t(\xi)} = \Phi(\widehat{Z}^{W^\top d\bar{h}_0(\xi_1)}, \ldots, \widehat{Z}^{W^\top d\bar{h}_0(\xi_m)}, \ldots, \widehat{Z}^{W^\top d\bar{h}_{t-1}(\xi_1)}, \ldots, \widehat{Z}^{W^\top d\bar{h}_{t-1}(\xi_m)}, Z^{U_0})$$

for some function $\Phi : \mathbb{R}^{m \times t+1} \to \mathbb{R}$. Then

$$\theta_{r,j} = \mathbb{E}[\partial \Phi(\widehat{Z}^{W^\top d\bar{h}_0(\xi_1)}, \ldots, \widehat{Z}^{W^\top d\bar{h}_0(\xi_m)}, \ldots, \widehat{Z}^{W^\top d\bar{h}_{t-1}(\xi_1)}, \ldots, \widehat{Z}^{W^\top d\bar{h}_{t-1}(\xi_m)}, Z^{U_0}) / \partial \widehat{Z}^{W^\top d\bar{h}_r(\xi_j)}].$$

## C. Proof of Theorem 4.5

We begin by describing three key lemmas, each highlighting a crucial aspect of our subsequent proof.

**Lemma C.1.** Suppose random variables $\{u_k\}_{k=[K]}$ and $\{v_k\}_{k=[K]}$ satisfy $\mathbb{E}[u_i u_j] = \mathbb{E}[v_i v_j], \forall i, j$, then

$$\sum_{k \in [K]} \alpha_k u_k \overset{a.s.}{=} 0 \Leftrightarrow \sum_{k \in [K]} \alpha_k v_k \overset{a.s.}{=} 0$$

*Proof.* $\sum_{k \in [K]} \alpha_k u_k \overset{a.s.}{=} 0$ implies $\mathbb{E}[(\sum_{k \in [K]} \alpha_k u_k)^2] = 0$. Because $\{u_k\}_{k=[K]}$ and $\{v_k\}_{k=[K]}$ share the same covariance matrix, we have that $\mathbb{E}[(\sum_{k \in [K]} \alpha_k v_k)^2] = 0$. □

**Lemma C.2.** Suppose any level set of $\phi : \mathbb{R} \to \mathbb{R}$ is countable and $g_1, \ldots, g_K$ are jointly non-degenerate Gaussian. If $\mathbb{P}(\sum_i a_i \phi(g_i) = C) > 0$ where $C$ is a constant, then $a_i = 0$ for all $i$ and $C = 0$.

*Proof.* $\{\mathbb{P}(\sum_i a_i \phi(g_i) = C | g_2, \ldots, g_K) > 0\}$ has positive probability only if $a_1 = 0$, because $g_1 | g_2, \ldots, g_K$ is a non-degenerate Gaussian random variable. We conclude that $\prod_{i \in [K]} a_i = 0$ following a similar reasoning inductively. □

**Lemma C.3.** Suppose $\phi$ satisfies Assumption 4.3. Moreover, suppose $g_1, \ldots, g_K$ are jointly non-degenerate Gaussian. If $(c_1 + \sum_i a_i \phi(g_i)) \cdot (c_2 + \sum_i b_i \phi'(g_i)) = C$ where $c_1, c_2, C$ is a constant, then $C = 0$ and either $a_i = 0$ for all $i \in [K]$ or $b_i = 0$ for all $i \in [K]$.

**Remark C.4.** Considering the function tail, it is easy to prove that the Sigmoid function $\sigma(x) = \frac{1}{1+\exp(-x)}$, the smoothed ReLU function $\overline{\text{ReLU}}(x) = \log(1 + \exp(x)) = \int \sigma(x) dx$, and the SiLU (Sigmoid Linear Unit) function, defined as $\text{SiLU}(x) = x \cdot \sigma(x)$, all satisfy these assumptions. Notably, SiLU is employed in state-of-the-art open-source foundation models (Touvron et al., 2023a;b).

*Proof.* We first prove that $C = 0$. Condition on the random variables $g_2, \ldots, g_K$ and denote $g := g_1 \mid (g_2, \ldots, g_K)$. Then $g$ is a non-degenerate univariate Gaussian.

**Case 1:** Suppose $C \neq 0$. - In this scenario, $a_1, b_1$ cannot be zero; otherwise $\phi$ (or $\phi'$) would have an uncountable level set, contradicting our assumptions. Given that $(c_1' + a_1 \phi(g))(c_2' + b_1 \phi'(g)) = C$ almost surely, we may rewrite:

$$\left( \frac{c_1'}{a_1} + \phi(g) \right) \left( \frac{c_2'}{b_1} + \phi'(g) \right) = \frac{C}{a_1 b_1}.$$

Here $c_1', c_2'$ are constants (absorbing the conditioning on $g_2, \ldots, g_K$). But this implies that for almost all $x \in \mathbb{R}$, $\left( \frac{c_1'}{a_1} + \phi(x) \right) \left( \frac{c_2'}{b_1} + \phi'(x) \right) = \frac{C}{a_1 b_1}$, a contradiction to the assumption on the activation function. Therefore, a contradiction arises, implying $C$ must be zero.

**Case 2:** Now consider $C = 0$. Then at least one of the following holds with positive probability:

$$c_1 + \sum_i a_i \phi(g_i) = 0 \quad \text{or} \quad c_2 + \sum_i b_i \phi'(g_i) = 0.$$

In either case, applying Lemma C.2 (which crucially uses the fact that $\phi$ has at most countable level sets, forcing the sum to avoid being constant on any uncountable domain with positive probability unless all involved coefficients vanish) completes the proof of zeroing out the corresponding coefficients. Concretely, if

$$\mathbb{P}\left( \sum_i a_i \phi(g_i) = C \Big| g_2, \ldots, g_K \right) > 0$$

then for those realizations we view $g_1$ (conditioned on $g_2, \ldots, g_K$) as a non-degenerate univariate Gaussian. Holding $g_2, \ldots, g_K$ fixed, the only way $\sum_i a_i \phi(g_i)$ can remain a constant over a positive-measure set of $g_1$ values is if $a_1 = 0$. Repeating this argument inductively for $g_2, g_3, \ldots$ shows that $\prod_{i \in [K]} a_i = 0$. Therefore, either all $a_i$ vanish or all $b_i$ vanish, completing the proof of this lemma. $\square$

With these lemmas at hand, we now prove our main theorem by an inductive argument. In particular, we show that the following two families of Gaussian processes, introduced in Section 4, remain non-degenerate throughout training:

$$\{\widehat{Z}^{W_0^l \delta x_s^{l-1}}(\xi_i)\}_{i \in [m], s \in [t], 2 \leq l \leq L}, \tag{C.1}$$

$$\{\widehat{Z}^{W_0^{l\top} dh_s^l}(\xi_i)\}_{i \in [m], s \in [t], 2 \leq l \leq L}. \tag{C.2}$$

Recall that a Gaussian process is non-degenerate if its covariance matrix $C$ at any finite collection of points satisfies $\det(C) \neq 0$ (Adler & Taylor, 2009). Using the filtration framework introduced in Section 4, our proof follows the natural flow of computation in the network, proceeding layer by layer and separately handling forward and backward passes. We break this into four key steps, each building upon the results of previous steps:

- Step 1: prove non-degeneracy for the features in the first hidden layer $\widehat{Z}^{W_0^2 \delta x_s^1}(\xi_i)$. This forms our base case as it only depends on the input data and network initialization, providing the foundation for our inductive argument.

- Step 2: prove non-degeneracy for the features in remaining layers $\widehat{Z}^{W_0^l \delta x_s^{l-1}}(\xi_i)$, $3 \leq l \leq L$. This step leverages the non-degeneracy established in Step 1 and shows how it propagates through deeper layers of the network.

- Step 3: prove non-degeneracy for the gradients in the last layer $\widehat{Z}^{W_0^{L\top} dh_s^L}(\xi_i)$. Here we transition from analyzing forward features to backward gradients, showing how the established feature properties ensure meaningful gradient flow.

- Step 4: prove non-degeneracy for the gradients in remaining layers $\widehat{Z}^{W_0^{l\top} dh_s^l}(\xi_i)$, $2 \leq l \leq L - 1$. Finally, we complete our analysis by showing how gradient non-degeneracy propagates backward through the network, ensuring effective training dynamics at all layers.

The proof proceeds by induction on the time step $t$, where at each step we verify these properties hold across all layers. This structure allows us to carefully track how the non-degeneracy property is maintained as information flows both forward and backward through the network during training. This systematic proof structure allows us to establish the global property of non-degeneracy by carefully tracking local changes at each layer and time step. We now proceed with the detailed proof.

*Proof of Theorem 4.5.* **Considering Trajectory Until Error Signals Vanish.** Throughout this proof, we focus on the training trajectory up to the time when all error signals $\mathring{\chi}_{t,i}$ become zero. This is because once the error signals vanish, there are no further parameter updates, and the training dynamics remain static thereafter. Our analysis ensures that up to this point, the Gaussian processes governing the feature and gradient updates remain non-degenerate, thereby maintaining the linear independence of features across all layers.

**Connecting $\widehat{Z}^{W\delta x}$ to $h^l$ and $x^l$.** Recall from Section 3 that each pre-activation $h^l(\xi)$ and post-activation $x^l(\xi)$ can be decomposed into a primary Gaussian increment plus lower-order (history-dependent) terms in the infinite-width limit:

Because these additional terms do not alter the essential covariance structure when conditioned on past information (they vanish or become deterministic in the limit), the linear (in)dependence of $\{h^l(\xi)\}$ or $\{x^l(\xi)\}$ is governed by the non-degeneracy of $\{\widehat{Z}^{W_0^l \delta x_s^{l-1}}(\xi)\}$. Hence, showing that $\{\widehat{Z}^{W_0^l \delta x_s^{l-1}}(\xi)\}$ remain non-degenerate under conditioning on historical variables directly implies that $\{h^l(\xi)\}$ and $\{x^l(\xi)\}$ cannot collapse into a linearly dependent set.

Below, we provide an inductive argument to establish precisely this non-degeneracy at each step.

By definition when $t = 0$, $\{\widehat{Z}^{W_0^l \delta x_0^{l-1}}(\xi_i)\}, i \in [m], 2 \leq l \leq L$ are independent and therefore non-degenerate Gaussian.

Now assume that the random Gaussian Process features defined in (C.1) and (C.2) are non-degenerate at time $t$, specifically for

$$\{\widehat{Z}^{W_0^l \delta x_s^{l-1}}(\xi_i)\}_{i \in [m], s \in [t]}, \{\widehat{Z}^{W_0^{l\top} dh_s^l}(\xi_i)\}_{i \in [m], s \in [t]}$$

where layer $2 \leq l \leq L$.

**Step 1:** We first prove $\{\widehat{Z}^{W_0^2 \delta x_s^1}(\xi_i)\}_{i \in [m], s \in [t+1]}$ is non-degenerate. Suppose there exists not all zero $\{\lambda_{i,s}\}_{i \in [m], s \in [t+1]}$ such that

$$\sum_{i \in [m], s \in [t+1]} \lambda_{i,s} \widehat{Z}^{W_0^2 \delta x_s^1}(\xi_i) \overset{\text{a.s.}}{=} 0.$$

Since $\{\widehat{Z}^{W_0^2 \delta x_s^1}(\xi_i)\}_{i \in [m], s \in [t]}$ are non-degenerate, we conclude that $\{\lambda_{i,t+1}\}_{i \in [m]}$ are not all zero. Consider the second moment, we have that

$$\mathbb{E}\left[\left(\sum_{i \in [m], s \in [t+1]} \lambda_{i,s} \widehat{Z}^{W_0^2 \delta x_s^1}(\xi_i)\right)^2\right] = 0.$$

Because $\{\widehat{Z}^{W_0^2 \delta x_s^1}(\xi_i)\}$ shares the same co-variance matrix with $\{Z^{\delta x_s^1}(\xi_i)\}$. By Lemma C.1, we have that

$$\mathbb{E}\left[\left(\sum_{i \in [m], s \in [t+1]} \lambda_{i,s} Z^{\delta x_s^1}(\xi_i)\right)^2\right] = 0 \Rightarrow \sum_{i \in [m], s \in [t+1]} \lambda_{i,s} Z^{\delta x_s^1}(\xi_i) \overset{a.s.}{=} 0.$$

Therefore by definition we have that

$$\sum_{i \in [m], s \in [t+1]} \lambda_{i,s}\left(\phi(Z^{h_s^1}(\xi_i)) - \phi(Z^{h_{s-1}^1}(\xi_i))\right) \overset{a.s.}{=} 0, \tag{C.3}$$

where $Z^{h_s^1}(\xi_i)$ satisfies that

$$Z^{h_s^1}(\xi_i) = Z^{h_0^1}(\xi_i) - \sum_{j \in [m]} \eta \mathring{\chi}_{0,j} \xi_j^\top \xi_i Z^{dh_0^1}(\xi_j) - \cdots - \sum_{j \in [m]} \eta \mathring{\chi}_{s-1,j} \xi_j^\top \xi_i Z^{dh_{s-1}^1}(\xi_j).$$

Plugging (3.6) and (3.7) into the above equation further gives the following reformulation of $Z^{h_s^1}(\xi_i)$:

$$Z^{h_s^1}(\xi_i) = \Delta_s(\xi_i) - \sum_{j \in [m]} \eta \mathring{\chi}_{s,j} \xi_j^\top \xi_i \phi'(Z^{h_{s-1}^1}(\xi_j)) \widehat{Z}^{W_0^{2\top} dh_s^2}(\xi_j), \tag{C.4}$$

where $\Delta_s(\xi_i) \in \mathcal{G}_{s-1}$ is a random variable. Notice that $Z^{h_{s-1}^1}(\xi_j) \in \mathcal{G}_{s-1}$ and $Z^{h_s^1}(\xi_j) \in \mathcal{G}_s$.

At least one of the $\{\mathring{\chi}_{s,j}\}_{j\in[m]}$ is not zero, W.L.O.G assume $\mathring{\chi}_{s,k} \neq 0$. By induction hypothesis the non-degenerate property at time $t$ holds. Therefore, $Z^{W_0^{2\top} dh_s^2(\xi_k)}$ condition on $\mathcal{G}_{t-1} \cup \{Z^{W_0^{2\top} dh_s^2(\xi_j)}\}_{j\neq k}$ is a non-degenerate Gaussian.

Plugging (C.4) into (C.3) and then condition on $\mathcal{G}_{t-1} \cup \{Z^{W_0^{2\top} dh_s^2(\xi_j)}\}_{j\neq k}$ gives that

$$\sum_{i\in[m]} \lambda_{i,t+1}\phi(\xi_k^\top \xi_i Z^U + c_i) = C$$

where $c_i$ and $C$ are constant and $Z^U$ is a non-degenerate uni-variate Gaussian random variable. Since $\phi$ meets the conditions in Assumption 4.3 and the dataset fulfills Assumption 4.1, ensuring the inner products and level sets behave as required. We can conclude that $\lambda_{i,t+1} = 0$ for all $i \in [m]$. A contradiction! Therefore, $\{\widehat{Z}^{W_0^2 \delta x_s^1(\xi_i)}\}_{i\in[m],s\in[t+1]}$ is indeed non-degenerate.

**Step 2:** We prove the following is non-degenerate.

$$\{\widehat{Z}^{W_0^l \delta x_s^{l-1}(\xi_i)}\}_{i\in[m],s\in[t+1]}, l \geq 3.$$

Suppose there exists not all zero $\{\lambda_{i,s}\}_{i\in[m],s\in[t+1]}$ such that

$$\sum_{i\in[m],s\in[t+1]} \lambda_{i,s}\widehat{Z}^{W_0^l \delta x_s^{l-1}(\xi_i)} \stackrel{\text{a.s.}}{=} 0.$$

Since $\{\widehat{Z}^{W_0^l \delta x_s^{l-1}(\xi_i)}\}_{i\in[m],s\in[t]}$ are non-degenerate, we conclude that $\{\lambda_{i,t+1}\}_{i\in[m]}$ are not all zero. Consider the second moment, we have that

$$\mathbb{E}\left[\left(\sum_{i\in[m],s\in[t+1]} \lambda_{i,s}\widehat{Z}^{W_0^l \delta x_s^{l-1}(\xi_i)}\right)^2\right] = 0.$$

Because $\{\widehat{Z}^{W_0^l \delta x_s^{l-1}(\xi_i)}\}$ shares the same co-variance matrix with $\{Z^{\delta x_s^{l-1}(\xi_i)}\}$. By Lemma C.1, we have that

$$\mathbb{E}\left[\left(\sum_{i\in[m],s\in[t+1]} \lambda_{i,s} Z^{\delta x_s^{l-1}(\xi_i)}\right)^2\right] = 0 \Rightarrow \sum_{i\in[m],s\in[t+1]} \lambda_{i,s} Z^{\delta x_s^{l-1}(\xi_i)} \stackrel{a.s.}{=} 0.$$

Therefore by definition we have that

$$\sum_{i\in[m],s\in[t+1]} \lambda_{i,s}\left[\phi(Z^{h_s^{l-1}(\xi_i)}) - \phi(Z^{h_{s-1}^{l-1}(\xi_i)})\right] \stackrel{a.s.}{=} 0, \tag{C.5}$$

where $Z^{h_s^{l-1}}$ satisfies that

$$Z^{h_s^{l-1}(\xi_i)} = Z^{h_0^{l-1}(\xi_i)} + Z^{\delta h_1^{l-1}(\xi_i)} + \cdots + Z^{\delta h_s^{l-1}(\xi_i)}.$$

A reformulation of the above update rule further gives that,

$$Z^{h_s^{l-1}(\xi_i)} = \Delta_s(\xi_i) + \widehat{Z}^{W_0^{l-1} \delta x_s^{l-2}(\xi_i)}, \tag{C.6}$$

where $\Delta_s(\xi_i) \in \mathcal{F}_{s-1}$ is a random variable. Notice that $\widehat{Z}^{W_0^{l-1} \delta x_s^{l-2}(\xi_i)}, Z^{h_s^{l-1}(\xi_i)} \in \mathcal{F}_s$. Arbitrary pick an index $k$. Because in induction hypothesis we assume the non-degenerate property at time $t$ for all layers and already proved the non-degenerate property at time $t+1$ layer $l-1$ , condition (C.3) on $\sigma\left(\mathcal{F}_t \cup \{\widehat{Z}^{W_0^{l-1} \delta x_{t+1}^{l-2}(\xi_j)}\}_{j\neq k}\right)$ gives that

$$\lambda_{k,t+1}\phi(Z^{U_k} + c_k) = C_k$$

where $U_k$ is a non-degenerate uni-variate Gaussian random variable $\widehat{Z}^{W_0^{l-1} \delta x_{t+1}^{l-2}(\xi_k)}|\sigma\left(\mathcal{F}_t \cup \{\widehat{Z}^{W_0^{l-1} \delta x_{t+1}^{l-2}(\xi_j)}\}_{j\neq k}\right)$, $c_k$ and $C_k$ are constants. By Assumption 4.3 of activation function, we know that $\lambda_{k,t+1} = 0$ for arbitrary $k \in [m]$. A contradiction! Therefore, $\{\widehat{Z}^{W_0^l \delta x_s^{l-1}(\xi_i)}\}_{i\in[m],s\in[t+1]}$ is indeed non-degenerate.

**Step 3:** We prove the following gradients are non-degenerate.

$$\{\widehat{Z}^{W_0^{L\top} dh_s^L(\xi_i)}\}_{i\in[m],s\in[t+1]}.$$

Suppose there exists not all zero $\{\lambda_{i,s}\}_{i\in[m],s\in[t+1]}$ such that

$$\sum_{i\in[m],s\in[t+1]} \lambda_{i,s}\widehat{Z}^{W_0^{L\top} dh_s^L(\xi_i)} \overset{\text{a.s.}}{=} 0.$$

Since $\{\widehat{Z}^{W_0^{L\top} dh_s^L(\xi_i)}\}_{i\in[m],s\in[t]}$ are non-degenerate, we conclude that $\{\lambda_{i,t+1}\}_{i\in[m]}$ are not all zero. Consider the second moment, we have that

$$\mathbb{E}\left[\left(\sum_{i\in[m],s\in[t+1]} \lambda_{i,s}\widehat{Z}^{W_0^{L\top} dh_s^L(\xi_i)}\right)^2\right] = 0.$$

Because $\{\widehat{Z}^{W_0^{L\top} dh_s^L(\xi_i)}\}$ shares the same co-variance matrix with $\{Z^{dh_s^L(\xi_i)}\}$. By Lemma C.1, we have that

$$\mathbb{E}\left[\left(\sum_{i\in[m],s\in[t+1]} \lambda_{i,s}Z^{dh_s^L(\xi_i)}\right)^2\right] = 0 \Rightarrow \sum_{i\in[m],s\in[t+1]} \lambda_{i,s}Z^{dh_s^L(\xi_i)} \overset{a.s.}{=} 0.$$

Therefore by definition we have that

$$\sum_{i\in[m],s\in[t+1]} \lambda_{i,s}Z^{dx_s^L(\xi_i)}\phi'(Z^{h_s^L(\xi_i)}) \overset{a.s.}{=} 0 \tag{C.7}$$

where $Z^{dx_s^L(\xi)}$ satisfies that

$$Z^{dx_s^L(\xi)} = Z^{\widehat{W}_s^{L+1}} = Z^{\widehat{W}_0^{L+1}} - \eta\sum_{s'=0}^{s-1}\sum_{i\in[m]} \mathring{\chi}_{s',i}Z^{x_{s'}^L(\xi_i)}$$

A reformulation of the above update rule further gives that,

$$Z^{dx_s^L(\xi)} = \widetilde{\Delta}_s - \eta\sum_{i\in[m]} \mathring{\chi}_{s,i}\phi(Z^{h_s^L(\xi_i)}),$$

$$Z^{h_s^L(\xi_i)} \overset{(i)}{=} \Delta_s + \widehat{Z}^{W_0^L \delta x_s^{L-1}(\xi_i)},$$

where $\Delta_s, \widetilde{\Delta}_s \in \mathcal{F}_s$ and (i) is due to (C.6). Notice that $Z^{dx_s^L}, Z^{h_s^L(\xi_i)} \in \mathcal{F}_s$. In (C.7), only $\widehat{Z}^{W_0^L \delta x_{t+1}^{L-1}(\xi_i)} \in \mathcal{F}_{t+1}$ provides new randomness. Because in induction hypothesis we assume the non-degenerate property at time $t$ for all layers and already proved the non-degenerate property of $\widehat{Z}^{W_0^L \delta x_{t+1}^{L-1}(\xi_i)}$, condition (C.7) on $\mathcal{F}_t$ gives that

$$\left(C - \eta\sum_{i\in[m]} \mathring{\chi}_{t,i}\phi(Z^{U_i} + b_i)\right) \cdot \left(\sum_{i\in[m],s\in[t+1]} \lambda_{i,t+1}\phi'(Z^{U_i} + b_i)\right) = C',$$

where $U_i = \widehat{Z}^{W_0^L \delta x_{t+1}^{L-1}(\xi_i)}|\mathcal{F}_t$ and $b_i, C, C'$ are all constants. Since $\mathring{\chi}_{t,i}$ are not all zero, by Lemma C.3, we have that $\lambda_{i,t+1} = 0$ for all $i \in [m]$. A contradiction! Therefore, $\{\widehat{Z}^{W_0^{L\top} dh_s^L(\xi_i)}\}_{i\in[m],s\in[t+1]}$ is indeed non-degenerate.

**Step 4:** We prove the following gradients are non-degenerate.

$$\{\widehat{Z}^{W_0^{l\top} dh_s^l(\xi_i)}\}_{i\in[m],s\in[t+1]}, 2 \leq l \leq L-1.$$

Suppose there exists not all zero $\{\lambda_{i,s}\}_{i\in[m],s\in[t+1]}$ such that

$$\sum_{i\in[m],s\in[t+1]} \lambda_{i,s}\widehat{Z}^{W_0^{l\top} dh_s^l(\xi_i)} \overset{\text{a.s.}}{=} 0.$$

Since $\{\widehat{Z}^{W_0^{l\top} dh_s^l(\xi_i)}\}_{i\in[m], s\in[t]}$ are non-degenerate, we conclude that $\{\lambda_{i,t+1}\}_{i\in[m]}$ are not all zero. Consider the second moment, we have that

$$\mathbb{E}\left[\left(\sum_{i\in[m],s\in[t+1]} \lambda_{i,s}\widehat{Z}^{W_0^{l\top} dh_s^l(\xi_i)}\right)^2\right] = 0.$$

Because $\{\widehat{Z}^{W_0^{l\top} dh_s^l(\xi_i)}\}$ shares the same co-variance matrix with $\{Z^{dh_s^l(\xi_i)}\}$. By Lemma C.1, we have that

$$\mathbb{E}\left[\left(\sum_{i\in[m],s\in[t+1]} \lambda_{i,s} Z^{dh_s^l(\xi_i)}\right)^2\right] = 0 \Rightarrow \sum_{i\in[m],s\in[t+1]} \lambda_{i,s} Z^{dh_s^l(\xi_i)} \stackrel{a.s.}{=} 0.$$

Therefore by definition we have that

$$\sum_{i\in[m],s\in[t+1]} \lambda_{i,s} Z^{dx_s^l(\xi_i)} \phi'(Z^{h_s^l(\xi_i)}) \stackrel{a.s.}{=} 0, \tag{C.8}$$

where $Z^{dx_s^l(\xi)}$ satisfies that

$$Z^{dx_s^l(\xi_i)} = \widehat{Z}^{W_0^{l+1\top} dh_s^{l+1}(\xi_i)} + G_s(\xi_i), \tag{C.9}$$

Similar to Steps 1 and 2, we have that $Z^{dx_s^l(\xi_i)} \in \mathcal{G}_s$, $Z^{h_s^l(\xi_i)} \in \mathcal{G}_{s-1}$. Therefore, condition (C.8) on $\mathcal{G}_t$ only $\widehat{Z}^{W_0^{l+1\top} dh_{t+1}^{l+1}(\xi)}$ gives new randomness. Arbitrarily pick an index $j$. Because in induction hypothesis we assume the non-degenerate property at time $t$ for all layers and already proved the non-degenerate property at time $t+1$ layer $l+1$, condition (C.8) on $\mathcal{G}_t \cup \{\widehat{Z}^{W_0^{l+1\top} dh_{t+1}^{l+1}(\xi_i)}\}_{i\neq j}$ gives that

$$\lambda_{j,t+1} c_j Z^{U_j} = C_j$$

where $U_j$ is a non-degenerate uni-variate Gaussian random variable $c_j$ and $C_j$ are constants. By Assumption 4.3 of activation function, we know that $c_j \neq 0$ which induces $\lambda_{j,t+1} = 0$ for all $j \in [m]$. A contradiction! Therefore, $\{\widehat{Z}^{W_0^{l\top} dh_s^l(\xi_i)}\}_{i\in[m], s\in[t+1]}$ is indeed non-degenerate.

$\square$

## Proof of Corollary 4.6

*Proof.* As stated in the main text, if the training parameters stop updating at time $T$, then the training loss must be zero.

By Theorem 4.5, the training trajectory remains non-degenerate throughout training. Suppose, for contradiction, that at time $T$ the training loss is still nonzero for some sample $(\xi_i, y_i)$. This implies that the error signal $\mathring{\chi}_{T,i}$ is nonzero. However, the non-degenerate trajectory ensures that a nonzero error signal $\mathring{\chi}_{T,i}$ would necessitate further parameter updates.

Specifically, we establish this through a detailed contradiction argument. Suppose at time $T$, there exists some sample $i$ with non-zero error signal $\mathring{\chi}_{T,i} \neq 0$, yet the parameters no longer update after time $T$.

According to our parameter update rule in (3.3), for the parameters to remain unchanged from time $T$ to $T+1$, we have:

$$Z_{\delta W_t^{L+1}} = -\eta \sum_{i\in[m]} \mathring{\chi}_{t-1,i} Z_{t-1}^{x^L}(\xi_i),$$

where $Z_{\delta W_t^{L+1}}$ represents the weight update at step $t$, $[m] = \{1, 2, ..., m\}$ denotes the set of indices for the training samples, and as stated in Section 3, the weights evolve as:

$$Z_{W_t^{L+1}} = Z_{W_0^{L+1}} + Z_{\delta W_1^{L+1}} + \cdots + Z_{\delta W_t^{L+1}}.$$

For the parameters to remain unchanged from time $T$ to $T+1$, we must have: $Z_{\delta W_{T+1}^{L+1}} = 0$. Substituting the update rule, this implies:

$$-\eta \sum_{i\in\mathcal{B}_T} \mathring{\chi}_{T,i} Z_T^{x^L}(\xi_i) = 0.$$

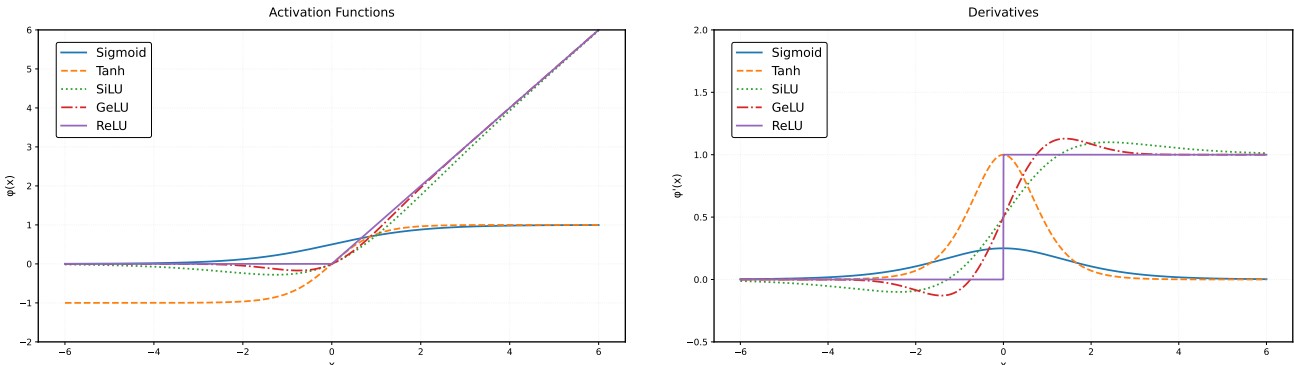

*Figure 17.* Different activation functions (left) and their derivatives (right). Note the exponential decay behavior in the tails of $\sigma$, tanh, SiLU, and GeLU, which is crucial for the GOOD property.

Since the learning rate $\eta > 0$, this simplifies to: $\sum_{i \in \mathcal{B}_T} \mathring{\chi}_{T,i} Z_T^{x^L}(\xi_i) = 0$. However, by Theorem 4.5, we have established that the post-activation features $Z_T^{x^L}(\xi_i)_{i \in [m]}$ are linearly independent at any time $T$. Since $\mathcal{B}_T \subseteq [m]$ and the features $\{Z_T^{x^L}(\xi_i)\}_{i \in [m]}$ are linearly independent, any subset of these features is also linearly independent. Therefore, the only way for the equation $\sum_{i \in \mathcal{B}_T} \mathring{\chi}_{T,i} Z_T^{x^L}(\xi_i) = 0$ to hold is if $\mathring{\chi}_{T,i} = 0$ for all $i \in \mathcal{B}_T$. This contradicts our assumption that $\mathring{\chi}_{T,i} \neq 0$ for some $i$. Therefore, if any error signal is non-zero at time $T$, the parameters must continue to update.

Moreover, since this argument applies to any batch $\mathcal{B}_t$ for $t \geq T$, we can conclude that if the model converges at time $T$ (meaning parameters no longer update), then all error signals must be zero: $\mathring{\chi}_{T,i} = 0$ for all $i \in \bigcup_{t \geq T} \mathcal{B}_t$, implying convergence to a global minimum. $\qquad\square$

## D. Activation Functions with the GOOD Property

We now verify that many practical activation functions, especially those with *exponential tails*, satisfy the GOOD property introduced in Definition 4.2. By "exponential tail," we mean that as $|x| \to \infty$, the function and/or its derivatives decay at least as fast as $e^{-c|x|}$ for some $c > 0$. Representative examples include the **sigmoid**, **tanh**, **SiLU**, and **GeLU**. Below, we restate the full definition of GOOD and then show how each requirement is met by these exponential-tail activations.

**Definition D.1** (Restatement of Definition 4.2). An activation function $\phi : \mathbb{R} \to \mathbb{R}$ is called GOOD if it satisfies the following two conditions:

**(a) Non-constant decomposition.** For any finite set of parameters $\{a_i\}, \{b_i\}, \{c_i\}$ such that $\exists k$ with $a_k b_k \neq 0$ and $|b_i| \neq |b_j|$ for all $i \neq j$, the function

$$f(x) = \sum_{i=1}^{m} a_i \phi(b_i x + c_i) \tag{D.1}$$

is *not* a constant function.

**(b) Non-degenerate product with derivative.** For any real numbers $r_1, r_2$, the product

$$\big(r_1 + \phi(x)\big)\big(r_2 + \phi'(x)\big) \tag{D.2}$$

is *not* almost everywhere (a.e.) constant on $\mathbb{R}$.

Before analyzing each activation function in detail, we visualize these functions and their derivatives in Figure 17. These plots illustrate the key characteristics we will exploit in our proofs, particularly the exponential decay behavior in the tails. Note how most activation functions and their derivatives exhibit rapid decay as $|x| \to \infty$, with ReLU serving as a contrasting example that grows linearly.

In the following subsections, we formally prove that these exponential-tail activations satisfy both conditions of Definition D.1.

## D.1. Sigmoid and Tanh

**Proposition D.2.** The sigmoid function $\sigma(x) = \frac{1}{1+\exp(-x)}$ satisfies both (a) and (b) in Definition D.1, hence is GOOD.

*Proof.* We first prove condition (a). Without loss of generality, set $c_i = 0$, as they will not affect the tail of the activation function. Define $\Omega = \{i|a_i \neq 0\}$, $A^+ = \{i \in \Omega|b_i > 0\}$ and $A^- = \{i \in \Omega|b_i < 0\}$. Let $i^* = \operatorname{argmin}_{i \in \Omega} |b_i|$. If $b_{i^*} = 0$, we can redefine $\Omega \leftarrow \Omega \setminus \{i^*\}$ and $f \leftarrow f - a_{i^*}/2$ and reenter this proof. Thus we assume $b_{i^*} \neq 0$ without loss of generality.

We have:

$$f(x) = \sum_{i \in A^+} a_i \sigma(b_i x) + \sum_{i \in A^-} a_i \sigma(b_i x) = \sum_{i \in A^+} a_i - \sum_{i \in A^+} a_i[1 - \sigma(b_i x)] + \sum_{i \in A^-} a_i \sigma(b_i x).$$

For $b_{i^*} < 0$, we have:

$$\left| f(x) - \sum_{i \in A^+} a_i \right| = \left| \sum_{i \in A^+} a_i[1 - \sigma(b_i x)] - \sum_{i \in A^-} a_i \sigma(b_i x) \right| = |a_{i^*}| \sigma(b_{i^*} x) + O(\exp(-Bx)),$$

where $B > |b_{i^*}|$. This dominant term cannot be cancelled unless $a_{i^*} = 0$.

For $b_{i^*} > 0$:

$$\left| f(x) - \sum_{i \in A^+} a_i \right| = \left| \sum_{i \in A^+} a_i[1 - \sigma(b_i x)] - \sum_{i \in A^-} a_i \sigma(b_i x) \right| = |a_{i^*}|[1 - \sigma(b_{i^*} x)] + O(\exp(-Bx)),$$

where $B > |b_{i^*}|$. This shows $f(x)$ cannot be constant unless $a_{i^*} b_{i^*} = 0$, contradicting our assumption.

For condition (b), we need to show $(r_1 + \sigma(x))(r_2 + \sigma'(x))$ is not a.e. constant. Note $\sigma'(x) = \sigma(x)(1 - \sigma(x))$ has exponential decay as $|x| \to \infty$. A direct computation shows:

$$(r_1 + \sigma(x))(r_2 + \sigma'(x)) = r_1 r_2 + r_2 \sigma(x) + r_1 \sigma(x)(1 - \sigma(x)) = r_1 r_2 + (r_2 + r_1)\sigma(x) - r_1 \sigma(x)^2 \tag{D.3}$$

Consider the tail in (D.3). If this expression were constant, then by examining the coefficients of different powers of $\sigma(x)$, we must have $r_1 = 0$ and $r_1 + r_2 = 0$, which is impossible. Thus $(r_1 + \sigma(x))(r_2 + \sigma'(x))$ cannot be constant almost everywhere. $\square$

**Remark D.3.** Since $\tanh(x)$ is a linear transformation of Sigmoind function $\sigma$, it inherits the same exponential-tail property and similarly meets both (a) and (b).

## D.2. SiLU and GeLU

**Proposition D.4.** The SiLU function $\operatorname{SiLU}(x) = x\sigma(x)$ is GOOD.

*Proof.* Define $\Omega = \{i|a_i \neq 0\}$, $A^+ = \{i \in \Omega|b_i > 0\}$ and $A^- = \{i \in \Omega|b_i < 0\}$. Let $i^* = \operatorname{argmin}_{i \in \Omega} |b_i|$. Using similar reasoning as in the sigmoid case, we assume $b_{i^*} \neq 0$ without loss of generality.

We have:

$$f(x) = \sum_{i \in A^+} a_i \phi(b_i x) + \sum_{i \in A^-} a_i \phi(b_i x) = \sum_{i \in A^+} a_i b_i x - \sum_{i \in A^+} a_i[b_i x - \phi(b_i x)] + \sum_{i \in A^-} a_i \phi(b_i x).$$

For $b_{i^*} < 0$, we have:

$$\left| f(x) - \sum_{i \in A^+} a_i b_i x \right| = \left| \sum_{i \in A^+} a_i[b_i x - \phi(b_i x)] - \sum_{i \in A^-} a_i \phi(b_i x) \right| = \underbrace{|a_{i^*}| \phi(b_{i^*} x)}_{\text{Majortail}} + O(x \exp(-Bx)),$$

where $B > |b_{i^*}|$.

For $b_{i^*} > 0$, we have:

$$\left| f(x) - \sum_{i \in A^+} a_i b_i x \right| = \left| \sum_{i \in A^+} a_i[b_i x - \phi(b_i x)] - \sum_{i \in A^-} a_i \phi(b_i x) \right| = \underbrace{|a_{i^*}|[b_{i^*} x - \phi(b_{i^*} x)]}_{\text{Majortail}} + O(x \exp(-Bx)),$$

where $B > |b_{i^*}|$. Note that $\text{Majortail}$ is bounded by some constant and asymptotically $\text{Majortail} = \Theta(x \exp(-|b_{i^*}|x))$. Therefore, $f(x)$ is constant only if $\sum_{i \in A^+} a_i b_i = 0$ and $a_{i^*} b_{i^*} = 0$, which contradicts our assumption.

For condition (b), we need to show $(r_1 + x\sigma(x))(r_2 + \sigma(x) + x\sigma'(x))$ is not a.e. constant. Note that:

$$\phi'(x) = \sigma(x) + x\sigma'(x) = \sigma(x) + x\sigma(x)(1 - \sigma(x)) \tag{D.4}$$

Then we have:

$$\begin{aligned}
&(r_1 + x\sigma(x))(r_2 + \sigma(x) + x\sigma(x)(1 - \sigma(x))) \\
&= r_1 r_2 + r_1 \sigma(x) + r_1 x\sigma(x)(1 - \sigma(x)) \\
&\quad + r_2 x\sigma(x) + x\sigma(x)^2 + x^2\sigma(x)^2(1 - \sigma(x))
\end{aligned} \tag{D.5}$$

Consider the tail in (D.5). If this expression were constant, the coefficient of $x^2$ term must vanish, which requires $\sigma(x)^2(1 - \sigma(x)) \equiv 0$. However, this is impossible as $\sigma(x) \in (0, 1)$ for all $x$. Thus this product cannot be constant almost everywhere. $\square$

**Remark D.5. GeLU**, defined by $x\Phi(x)$ where $\Phi$ is the Gaussian CDF, similarly satisfies (a) and (b) because of its strong exponential decay. Specifically, as $|x| \to \infty$, GeLU and its derivatives exhibit Gaussian-like decay $O(e^{-x^2/2})$, which is even stronger than the exponential decay of sigmoid and SiLU.

**Conclusion.** We have shown that key exponential-tail activations ($\sigma$, $\tanh$, SiLU, GeLU) fulfill both (a) and (b) in Definition D.1, and hence are GOOD. These results rely crucially on the exponential decay properties of these functions, which ensure that scaled copies cannot combine to yield constant functions. This ensures rich, non-degenerate behavior in our infinite-width analysis under $\mu$P scaling.

