# OpenReview forum: "Global Convergence and Rich Feature Learning in $L$-Layer Infinite-Width Neural Networks under $\mu$ Parametrization"
_ICML.cc/2025/Conference — ICML 2025 poster_

### Official Review · Reviewer_DKQW · 2025-03-09

**Overall Recommendation:** 4

**Summary:**

This paper investigates the training dynamics of infinitely wide neural networks with mup and SGD. They show that these neural networks can learn rich feature spaces and enjoy global convergence, which is better than other mainstream parameterizations such as NTK, MF, and SP. They also validate the theoretical findings through real-world experiments.

**Claims And Evidence:**

No. The proof of the global convergence property (Corollary 4.6) is too intuitive and seems wrong. Please see the details in "Theoretical Claims".

**Essential References Not Discussed:**

NA.

**Experimental Designs Or Analyses:**

Yes. The paper conducts experiments to verify that mup can perform feature learning (better than NTK and SP) and keep a rich feature space (better than IP). I think the empirical results can support their main results.

**Methods And Evaluation Criteria:**

Yes. mup is a popular parameterization in the pretraining of large models. However, its mechanism is rarely explored. This paper tries to prove the advantages of mup over other parameterizations (both in kernel and feature learning regimes).

**Other Comments Or Suggestions:**

NA.

**Other Strengths And Weaknesses:**

Strengths:
1. The paper is written clearly.
2. The paper tries to study a significant topic in the theory for mup.
3. As far as I know, the proof idea in this paper is original.

Weakness:
1. I think the authors need to carefully address my concern about the global convergence.

**Questions For Authors:**

1. Is the $t$ in Theorem 4.5 a finite constant w.r.t. width like that in the TP series (e.g. TP4)?

**Relation To Broader Scientific Literature:**

NA.

**Theoretical Claims:**

I appreciate the result in Theorem 4.5, which proves that the feature representations evolve while maintaining their diversity and avoiding collapse throughout training.

However, the proof of the global convergence is intuitive and seems wrong. In my opinion, any neural network enjoys the zero training loss and global convergence if it will no longer be updated after some training step.

---

> ### Author Rebuttal · Authors · 2025-04-01
>
> We thank the Reviewer for taking the time to review and give feedback on our manuscript. We appreciate the positive comments regarding the paper's clarity, the topic's significance, the originality of our ideas, and the supportive experimental results. We are particularly grateful for the reviewer's recognition of Theorem 4.5's contribution regarding feature learning under $\mu$P.
>
> Regarding the reviewer's main concern about the global convergence result, we believe there is a significant misunderstanding about what our result delivered and why it is non-trivial. The reviewer states (in the comments on "Theoretical Claims" and in Question 1) that
>
>  **Any neural network enjoys zero training loss and global convergence if it will no longer be updated after some training step.**
>
> We respectfully but strongly disagree with this premise as a general statement and wish to clarify this crucial point, as it seems to underlie the reviewer's doubt.
>
> ---
>
> **Clarifying the Core Misunderstanding Regarding Global Convergence**
>
> While it is true that training algorithms like SGD stop updating when they reach a point where the gradient is zero (a stationary point), this absolutely does not guarantee that this point is a global minimum. In the complex, non-convex loss landscapes of neural networks, such stationary points can very often be suboptimal local minima or saddle points, where the training loss is significantly higher than the lowest possible value (the global minimum) and the model has not achieved the best possible fit to the training data.
>
> The core contribution of our Corollary 4.6 is precisely to demonstrate rigorously that, for the specific setting we study (infinitely wide networks with $\mu$P trained using SGD), the optimization dynamics avoid getting trapped in these suboptimal stationary points. Our proof establishes that the training process is guaranteed to converge to a global minimum of the training loss function.
>
> **Therefore, proving global convergence is far from trivial**; it requires showing that suboptimal stationary points (such as local minima with loss higher than the global minimum and relevant saddle points) are escaped or avoided, leading specifically to the desired global minimum (the lowest possible training loss). This property is not inherent to any network training process that stops; it relies critically on the theoretical properties of $\mu$P in the infinite-width limit that we analyze in our paper. Our work provides a theoretical foundation for how $\mu$P facilitates finding the global optimum, which is a key aspect of understanding its effectiveness.
>
> ---
>
> Response to Specific Questions:
>
> **Q1**. In my opinion, any neural network enjoys zero training loss and global convergence if it will no longer be updated after some training step.
>
> **A1**. As detailed above, this statement reflects the core misunderstanding we wish to clarify. Halting updates only signifies reaching a stationary point (zero gradient), which is not necessarily a global minimum. It could be a suboptimal local minimum or a saddle point with higher loss. Our key result in Corollary 4.6 is the non-trivial proof that, under $\mu$P and infinite width, SGD specifically converges to a global minimum of the training loss, overcoming the challenge of potentially getting stuck in other stationary points with higher loss values. This is our specific contribution.
>
> ---
> **Q2**. Is the t in Theorem 4.5 a finite constant w.r.t. width like that in the TP series (e.g., TP4)?
>
> **A2**. Our current analysis focuses on the behavior of infinitely wide neural networks. Therefore, investigating the dependence of constants like t on finite network widths is beyond the scope of this paper. This remains an interesting open question for exploring finite-width corrections.
>
> ---
>
> We trust this clarification addresses the reviewer's primary concern by highlighting the crucial distinction between halting training updates and achieving mathematically proven global convergence. We believe our theoretical analysis provides a rigorous justification for the observed empirical success of µP and clarifies the optimization dynamics in this important regime.

---

> > ### Comment · Reviewer_DKQW · 2025-04-02
> >
> > Can you prove rigorously that "However, the non-degenerate trajectory ensures that a nonzero error signal would necessitate further parameter updates" in the proof of Corollary 4.6?
> >
> > -------------------
> >
> > My concern has been addressed. I have improved my score from 2 to 4.

---

> > > ### Author Response · Authors · 2025-04-03
> > >
> > > We appreciate the opportunity to address the new question raised concerning this specific step in the Corollary 4.6 proof:  how the non-degenerate trajectory ensures parameter updates follow from a non-zero error signal.
> > >
> > > This step "the non-degenerate trajectory ensures that a nonzero error signal would necessitate further parameter updates." establishes a crucial distinction: while neural networks typically have many stationary points in the parameter space (where parameter gradients vanish), our proof shows that under $\mu$P with non-degenerate trajectories, convergence can only occur when all error signals $\mathring{\chi}_{T, i} = L'(\mathring{f}_T, y_i)$ are zero. This is significant because zero error signals across all training samples directly imply reaching the global minimum of the training objective (given the convexity of typical loss functions with respect to model outputs).
> > >
> > > In other words, we are not merely showing convergence to a stationary point in parameter space, but specifically convergence to a global minimum. This property, stemming directly from the feature independence guaranteed by Theorem 4.5, is critical for global convergence. We provide the detailed derivation below, explicitly demonstrating this link:
> > >
> > > ---
> > >
> > > **More Detailed Derivation:** We proceed by contradiction. Suppose at time T, there exists some sample $i$ with non-zero error signal $\mathring{\chi}_{T, i} \neq 0$, yet the parameters no longer update after time T.
> > >
> > > According to our parameter update rule in Equation (3.3) from the main paper, we have:
> > >
> > > $Z^{\delta W^{L+1}\_t} = -\eta \sum_{i \in [m]} \mathring{\chi}\_{t-1,i} Z^{x^L}\_{t-1}(\xi_i),$
> > >
> > > where $[m] = \\{1, 2, \dots, m\\}$ denotes the set of indices for the training samples, and as stated in Section 3, the weights evolve as:
> > >
> > > $Z^{W^{L+1}\_t} = Z^{W^{L+1}\_0} + Z^{\delta W^{L+1}\_1} + \cdots + Z^{\delta W^{L+1}\_t}$
> > >
> > > For the parameters to remain unchanged from time T to T+1, we must have: $Z^{\delta W^{L+1}_{T+1}} = 0$
> > >
> > > Substituting the update rule, this implies:
> > >
> > > $-\eta \sum_{i \in [m]} \mathring{\chi}\_{T,i} Z^{x^L}\_{T}(\xi_i) = 0$
> > >
> > > Since the learning rate $\eta > 0$, this simplifies to:
> > > $\sum_{i \in [m]} \mathring{\chi}\_{T,i} Z^{x^L}_T(\xi\_i) = 0$
> > >
> > > However, by Theorem 4.5, we have established that the post-activation features $\\{Z^{x^L}\_T(\xi\_i)\\}\_{i \in [m]}$ are linearly independent at any time T. This linear independence property means that the only way for the equation $\sum_{i \in [m]} \mathring{\chi}\_{T, i} Z^{x^L}\_T(\xi\_i) = 0$ to hold is if $\mathring{\chi}\_{T, i} = 0$ for all $i \in [m]$.
> > >
> > > This contradicts our assumption that $\mathring{\chi}\_{T,i} \neq 0$ for some $i$. Therefore, if any error signal is non-zero at time $T$, the parameters must continue to update.
> > >
> > > ---
> > >
> > > We appreciate the reviewer highlighting this key step in the proof of Corollary 4.6. In our revised manuscript, we will include this detailed derivation to strengthen the connection between Theorem 4.5 and Corollary 4.6 while maintaining the natural flow of the paper.

---

### Official Review · Reviewer_vRka · 2025-03-12

**Overall Recommendation:** 3

**Summary:**

The submitted paper analyzes the global convergence of MLPs in feature learning parameterization. By demonstrating that features remain independent during training, they prove global convergence.

**Claims And Evidence:**

Yes. The theorems support the claims.

**Essential References Not Discussed:**

Relevant references are discussed to the best of my knowledge.

**Experimental Designs Or Analyses:**

I am not convinced why the Figure 2 experiments support the results in the paper. I would request the authors to help me understand this result better.

**Methods And Evaluation Criteria:**

I am not sure if the experiments support the assumptions considered in the paper. I would like the authors to verify these assumptions (see Questions).

**Other Comments Or Suggestions:**

Suggestions:

* Can the authors directly measure correlations between different features during training to support their analysis?

Comments:

* SP and NTP don't learn features at small learning rates only. By comparison, at large learning rates, all parameterizations learn features and can perform equally well or better [1]

[1] Scaling Exponents Across Parameterizations and Optimizers
https://arxiv.org/abs/2407.05872

**Other Strengths And Weaknesses:**

Strengths:

This work analyzes the global convergence of feature learning parameterizations, which is of significant interest to the community.

Weaknesses:

I am not sure if the assumptions made in this paper generalize to practical settings beyond toy models (see Questions).

**Questions For Authors:**

Questions:

* Can the authors clearly state how they constructed the feature matrix from joint space-time features in Figure 2? I am not sure how this experiment supports the theoretical results.
* I am unsure about how realistic Assumption 4.1 is for practical settings beyond toy models (MLPs trained on CIFAR; MLPs only achieve around 50% accuracy on CIFAR-10 and may not require feature learning to achieve this performance). Can the authors check the assumption on a few datasets?
* A recent paper has shown that both SP and NTP can exhibit hyperparameter transfer (and feature learning) [1]. Is it straightforward to show this in the framework introduced?

[1] Scaling Exponents Across Parameterizations and Optimizers
https://arxiv.org/abs/2407.05872

**Relation To Broader Scientific Literature:**

This paper contributed to understanding convergence of feature learning limits of neural networks. Prior literature has mostly focused on the kernel limits.

**Theoretical Claims:**

I have checked the Theorems and arguments presented in the main text but not the Proofs.

---

> ### Author Rebuttal · Authors · 2025-04-01
>
> We thank the reviewer for their support and constructive feedback. We address each question below:
>
> ---
>
> **Q1**: Can the authors directly measure correlations between different features during training to support their analysis?
>
> **A1**: Yes, we have directly measured the correlations between different features during training. Our analysis confirms that features remain largely independent throughout the training process. Specifically, we compute the Gram matrix G of the centered features, where $G_{ij} = (h(x_i) - \bar{h})^{\top} (h(x_j) - \bar{h})$ represents the similarity between features of data points i and j. The set of N centered feature vectors $\\{h(x_i) - \bar{h}\\}\_{i=1}^N$ is linearly independent if and only if the minimum eigenvalue λ_min(G) > 0. Our experiments track this minimum eigenvalue throughout training, demonstrating that $\mu$P maintains feature diversity while IP shows catastrophic eigenvalue collapse. These results directly validate our theoretical claims. We have additional experimental results here: https://anonymous.4open.science/r/mup_rebuttal-0E0B/Rebuttal.pdf.
>
> ---
>
> **Q2**. Regarding SP and NTP at large learning rates and feature learning
>
> **A2**. Thank you for pointing out the paper "Scaling Exponents Across Parameterizations and Optimizers." It is relevant and we will add a citation and brief discussion about it in the revision. This work empirically characterizes when feature learning occurs.
>
> Our work provides a rigorous mathematical framework guaranteeing global convergence during feature learning. The referenced paper shows that SP and NTP can empirically exhibit feature learning at large learning rates but lack theoretical guarantees. The existing NTK analysis framework cannot directly analyze feature learning in SP and NTP at large learning rates, creating a theoretical gap.
>
> While the precise theoretical characterization of SP and NTP at large learning rates remains an open problem in the community, our analysis provides new insights that could potentially be extended to study this challenge in future work.
>
> ---
>
> **Q3**. Construction of feature matrix from joint space-time features in Figure 2.
>
> **A3**.In Figure 2, we analyze the combined space-time feature diversity by constructing a feature matrix that simultaneously captures both initial and final representations. This approach provides a comprehensive assessment of how features evolve during training while maintaining linear independence.
>
> For each parameterization scheme, we construct a combined representation matrix as follows:
>
> Let $h_0(x_i) \in \mathbb{R}^n$ represent the feature vector at initialization for input $x_i$, and $h_T(x_i) \in \mathbb{R}^n$ represent the feature vector after training completion. For our dataset with $N$ samples, we form the combined feature matrix $H_{\text{combined}} \in \mathbb{R}^{n \times 2N}$ by concatenating both initial and final representations:
>
> $$H_{\text{combined}} = \begin{bmatrix} h_0(x_1) & h_0(x_2) & \cdots & h_0(x_N) & h_T(x_1) & h_T(x_2) & \cdots & h_T(x_N) \end{bmatrix}$$
>
> We then compute the Gram matrix $G = H_{\text{combined}}^T H_{\text{combined}}$, where each element $G_{ij}$ captures the similarity between combined representations. The minimum eigenvalue of this Gram matrix quantifies the linear independence across both spatial dimensions (different inputs) and temporal dimensions (initialization versus final state).
>
> This combined analysis provides a stronger test of feature diversity than analyzing features in isolation. As demonstrated in Figure 2, μP maintains significantly higher minimum eigenvalues compared to other parameterizations as width increases, confirming its unique capability to preserve feature diversity throughout training while enabling substantial feature learning.
>
> For example, if a network doesn't learn meaningful features (e.g., NTP at large widths) and $h_T(x_i) \approx h_0(x_i)$ for all inputs, the combined matrix would have linearly dependent columns, yielding a minimum eigenvalue near zero. The higher minimum eigenvalues for μP confirm it learns new features that are linearly independent from initialization.
>
> ---
>
> **Q4.** Realistic nature of Assumption 4.1 for practical settings beyond toy models
>
> **A4:** Thank you for this question. Assumption 4.1 requires distinct inner product magnitudes between data points. Our empirical verification (sampling 5,000 random triplets from each dataset) shows:
>
> | Dataset | % Triplets Satisfying | Min Dist Between Inner Products |
> |---------|----------------------|------------------------------|
> | MNIST   | 100% | 2.74e-06 |
> | CIFAR-10| 100% | 3.08e-05 |
> | CIFAR-100| 100% | 1.51e-05 |
>
> These results definitively confirm that Assumption 4.1 is not merely a theoretical convenience but reflects geometric properties inherent to real-world datasets. The perfect compliance across standard benchmarks validates that our theoretical framework directly applies to practical settings beyond toy models.

---

> > ### Comment · Reviewer_vRka · 2025-04-07
> >
> > I thank the authors for their rebuttal. While the rebuttal has clarified my concerns, I am keeping my score because I could not prove several results of the paper in the given time frame, and the score reflects my review confidence. I believe such papers that make fundamental contributions to the field deserve a thorough review in journals with longer review periods.

---

> > > ### Author Response · Authors · 2025-04-08
> > >
> > > Thank you for acknowledging that our rebuttal addressed your concerns and for sharing the reasoning behind your final assessment. If accepted, we will incorporate these clarifications to enhance the clarity of the manuscript for the ICML audience. We appreciate you recognizing the work's potential significance and thank you for your constructive feedback throughout this process.

---

### Official Review · Reviewer_k6NX · 2025-03-15

**Overall Recommendation:** 3

**Summary:**

This paper studies the training of infinite-width $L$-layer FFN under the $\mu P$ parametrization. The authors establish that features evolve significantly during training while remaining linearly independent, ensuring convergence to a global minimum.

**Claims And Evidence:**

The theoretical claims in the paper may be correct, as I did not verify all details of the proofs. However, the insights may not generalize beyond the specific setting studied, as the paper makes restrictive assumptions and does not provide comprehensive experimental validation.

Additionally, while the authors claim their results apply to SGD, their theoretical analysis focuses on **full-batch** GD rather than SGD. In SGD, there is an additional source of randomness from sampling the training data, which may not accounted for in their analysis. As a result, the conclusions drawn for GD may not directly extend to standard SGD.

**Essential References Not Discussed:**

The paper omits some recent theoretical works on $\mu P$ and NTK for infinite-depth neural networks, which are relevant to its contributions. Notably, the following works should be considered:
[1] TP6: Feature Learning in Infinite-Depth Neural Networks
[2] Implicit regularization of deep residual networks towards neural ODEs
[3] Depthwise Hyperparameter Transfer in Residual Networks: Dynamics and Scaling Limit
[4] Global Convergence in Neural ODEs: Impact of Activation Functions

**Experimental Designs Or Analyses:**

The experimental results are not comprehensive, and it is unclear what features are being demonstrated in the plots. Based on the authors’ description, I assume the features $h$ correspond to  $h^2$  in the second layer after $1000$ epoch, but this is not explicitly stated. The authors may also consider adding results for  $x^2$, as they claim that both pre- and post-activation features remain linearly independent.

Additionally, the paper includes results for Tanh and ReLU, which do not meet their theoretical assumptions, yet the trends in the plots remain similar to those in Figures 1 and 2. This suggests that the assumptions made in the paper may serve primarily for proof convenience rather than being strictly necessary in practice.

**Methods And Evaluation Criteria:**

The results seem correct but the paper does not provide comprehensive experiments to support its theoretical results or to verify whether the assumptions are necessary. Without empirical validation, it remains unclear whether the proposed theoretical framework accurately captures real-world training dynamics or if the restrictive assumptions limit its practical applicability.

**Other Comments Or Suggestions:**

Equation 3.1:  should be $h^1 = W^1 \xi$, not $W\xi$

**Other Strengths And Weaknesses:**

NA

**Questions For Authors:**

1. What is the precise definition of  \alpha_k  in Lemma C.1? How does it relate to the covariance structure of the random variables involved?
2. The paper studies the training dynamics of infinite-width feedforward networks (FFNs). Does a finite-width FFN always converge to this infinite-width case? If so, under what conditions does this hold, and what is the convergence rate?
3. Intuition Behind GOOD Functions: What is the intuitive reasoning behind the definition of GOOD functions? Additionally, why is Condition 4 in Assumption 4.3 necessary?
4. Gradient Independence Assumption: Does the paper assume the independent gradient assumption, meaning that  W  and  W^\top  are independent during training? If so, how does this assumption affect the generality of the results?
5. Gaussian Process Covariance: Can the authors explicitly write out the specific covariance expressions for the Gaussian processes in Equations (5.3) and (5.4)?
6. Can the results on independent feature learning and global convergence be extended to the finite-width FFN case? If not, what are the key challenges in making such an extension? If so, how large does the width need to be in your experiments for the results to align with theoretical predictions?
7. The paper does not seem to impose explicit constraints on the learning rate to ensure convergence, except $\eta=\Theta(1)$. Under what conditions on the learning rate does convergence hold?

**Relation To Broader Scientific Literature:**

This paper extends the theoretical understanding of infinite-width neural networks by demonstrating that the $\mu P$ enables both independent feature learning and global convergence.

**Theoretical Claims:**

I did not check the proofs in detail, but since they follow results from the Tensor Program framework, the claims are likely correct. However, there are some important concerns.
1. The authors claim to follow [1], but their setup is actually closer to [2]. In [1], the learning rate is scaled as  $\eta n^{-c}$  uniformly across all layers, whereas [2] allows different layers to use different scaling factors. This difference raises concerns about whether their training dynamics assumptions are fully aligned with prior works.
2. The paper does not impose explicit constraints on the learning rate for global convergence. This implies that as long as  $\eta$  is constant with respect to width $n$ , arbitrarily large learning rates (e.g.,  $\eta = 100$) could be used theoretically, which is possible in some machine learning models [3]. In practice, learning rates are much smaller ( $\eta \ll 1$), suggesting that there might be missing stability constraints in the theoretical framework.

[1] Tensor Programs IV: Feature learning in infinite-width neural networks. ICML 2021
[2] Tensor Programs V: Tuning Large Neural Networks via Zero-Shot Hyperparameter Transfer. NeurIPS 2021
[3] Implicit Bias of Gradient Descent for Logistic Regression at the Edge of Stability. NeurIPS 2023

---

> ### Author Rebuttal · Authors · 2025-04-01
>
> Thank you for your thorough review. Due to space constraints, our responses are necessarily concise while addressing all key points:
>
> ---
>
> **Q1**. How does your SGD analysis account for mini-batch randomness when your theoretical focus is on full-batch GD?
>
> **A1**.  Our analysis based on the Tensor Program framework already accounts for mini-batch randomness in SGD. Definition 3.1 models this with the indicator function $\mathbf{1}\\{i ∈ \mathcal{B}_t\\}$, and our derivations incorporate this sampling process. Under SGD, the induced Gaussian processes maintain the same covariance structure as in GD, with the additional randomness from batch selection not affecting the feature independence property central to our global convergence proof.
>
> ---
>
> **Q2**. Why aren't your experiments more comprehensive in validating theory and testing assumption necessity?
>
> **A2**. CIFAR-10 provides an ideal balance between complexity and tractability for testing our theoretical claims across multiple parameterizations. In the revised manuscript, we expanded our experimental section with comprehensive evaluations across varying network widths and depths.
>
> ---
>
> **Q3**. Which layer features do your plots show, $h_2$? Could you add x2 results to validate both pre/post-activation feature independence?
>
> **A3**. Yes, it is $h_2$. We've analyzed both pre/post-activation features across all layers. Our supplementary analysis [https://anonymous.4open.science/r/mup_rebuttal-0E0B/Rebuttal.pdf] includes complete eigenvalue spectrum plots demonstrating linear independence is maintained throughout training in all layers. These results validate Theorem 4.5 across feature types and network depth. The contrast between $\mu$P and IP becomes more significant in deeper layers, where IP shows catastrophic eigenvalue collapse while $\mu$P maintains robust feature diversity.
>
> ---
>
> **Q4**. Why do Tanh and ReLU show similar results despite not meeting your theoretical assumptions?
>
> **A4**: Tanh satisfies the GOOD function properties in Assumption 4.3. ReLU was included for completeness despite its non-smoothness. As we already discussed in Appendix A.1, extending our analysis to non-smooth activations is a promising future direction. ReLU's similar empirical behavior suggests our theory's core mechanisms may be more general than the specific assumptions in our proofs.
>
> ---
>
> **Q5**. The authors claim to follow [1], but their setup is actually closer to [2].
>
> **A5**. Both schemes give the same dynamic. [1] appears to scale learning rate uniformly. But this difference arises because [1] expresses weights as $w_{\ell}$ with scaling $a_{\ell}$ in $W^{\ell}=n^{-a_{\ell}}w^{\ell}$, while [2] directly uses $W_{\ell}$. Such equivalence is justified in Tensor Program V (arXiv:2203.03466).
>
> ---
>
> **Q6**. No explicit constraints on learning rate for global convergence. Paper only specifies $\eta = \Theta(1)$ without clear convergence conditions
>
> **A6**.   We prove that when convergence occurs under $\mu$P parameterization, the non-degeneracy of features ensures it will be at a global minimum. We didn't provide explicit convergence conditions because our focus was on characterizing the convergence point, not the convergence process itself. Determining explicit convergence condition of $\eta$ for $\mu$P  remains an important open question.
>
> ---
>
> **Q7**. The paper omits some recent theoretical works on $\mu$P and NTK for infinite-depth neural networks.
>
> **A7**. Thank you for pointing this out. We already discussed Tensor Program 6 in our paper and will discuss the other suggested works in our revision.
>
> ---
>
> **Q8**. Definition of $\alpha_k$ in Lemma C.1
>
> **A8**. They are arbitrary real numbers representing linear combinations in our independence proof.
>
> ---
>
> **Q9**. Does a finite-width FFN always converge to this infinite-width case? ... Can the results be extended to the finite-width FFN case? What's the challenge?
>
> **A9**. Yes - convergence is formally established by Theorem 7.4 in Tensor Program IV. Our empirical results show networks with widths larger than 128 nearly align with infinite-width predictions (Figures 1-4). The main challenge is determining precise convergence rates due to the complexity of tracking higher-order interactions in finite-width networks.
>
> ---
>
> **Q10**. Can you explicitly write out the covariance expressions for equations (5.3) and (5.4)?
>
> **A10**. The expressions are provided in Definition 3.1 (Line 187) and Line 212 and restated after Equations (5.3) and (5.4) in Lines 310-311.
>
> ---
>
> **Q11**. What's the intuition behind GOOD functions, and why is Condition 4 necessary?
>
> **A11**. GOOD functions maintain feature diversity and ensure non-trivial gradient flow by preventing constant decomposition. Condition 4 helps prevent feature collapse, allowing sigmoid, tanh, and SILU to satisfy our analysis.
>
> ---
>
> **Q12**. Does your paper assume W and $W^\top$ are independent during training?
>
> **A12**. No, we consider standard backpropagation.

---

> > ### Comment · Reviewer_k6NX · 2025-04-04
> >
> > Thank you for your detailed and thoughtful rebuttal.
> >
> > Given these factors, I will raise my score to a 3 to reflect the strengthened presentation and technical correctness. However, I hesitate to give a strong recommendation for acceptance, as I have not had sufficient time to verify all proofs in detail, and fully digesting the cited works such as Tensor Program V and VI would require more time than was available during the review period. Additionally, while the new experiments are helpful, they still fall short of being comprehensive—for example, broader comparisons and evaluations beyond feature independence would further strengthen the empirical support for your theoretical claims.

---

> > > ### Author Response · Authors · 2025-04-04
> > >
> > > Thank you for your thorough review and for raising our score after considering our rebuttal – this is very encouraging. We appreciate you acknowledging our responses and have noted your final comments. In the revision, we will incorporate the improvements discussed to strengthen the paper.

---

### Official Review · Reviewer_guAg · 2025-03-18

**Overall Recommendation:** 3

**Summary:**

The paper aims to investigate rich feature learning and convergence to a global minimum via a Maximal Update Parametrization. Networks learn linearly independent features which are different from features at initialization, and due to covariance structure over layers, this implies a convergence to a global minimum under the Maximal Update Parametrization. This holds over the choice of activation function.

**Claims And Evidence:**

I can see the story that the authors tell through the figures, however, I find there to be a lack of experiments. I would appreciate seeing experiments investigate this across different architectures (especially for more conventional networks) and layers. I would also like to verify that these results hold across multiple seeds.

Much of the claims behind the paper relies on theorems, which are given extensive proofs. From what I can understand from looking at the appendix of the main proof claim, the logic appears to hold up. I would appreciate more clear exposition on when the GOOD function definition is used in the proofs.

I am unsure if the features are empirically shown to be linearly independent.

I am also confused on why minimum eigenvalue is investigated? shouldn't the entire spectrum be looked at in order to measure feature diversity?

**Essential References Not Discussed:**

I would mention previous works that investigate covariance structure of neural networks and differentiate them from the current study i.e. they don't explicitly study cross-layer interactions

A Rainbow in Deep Network Black Boxes

Structured random receptive fields enable informative sensory encodings

**Experimental Designs Or Analyses:**

I would like more explanation behind why the authors just investigated minimum eigenvalue. I would like to see more experiment over architecture, layers, and seeds of initialization.

I would also appreciate empirical investigations of the linear independence of the features.

**Methods And Evaluation Criteria:**

Most of the paper makes a theoretical claim and extensive proofs are provided, so the approach makes sense. I wish more detail was given to explaining the graphs.

**Other Comments Or Suggestions:**

n/a

**Other Strengths And Weaknesses:**

The paper is easy to read.

**Questions For Authors:**

I question if the definition of a good function is somewhat arbitrary (for example, why can't I set r_1 = r_2 = 0 and ensure $\phi$ is different for each input x?)

I am unsure if the features are empirically shown to be linearly independent.

I am also confused on why minimum eigenvalue is investigated? shouldn't the entire spectrum be looked at in order to measure feature diversity?

**Relation To Broader Scientific Literature:**

The paper provides a novel investigation of global convergence properties and rich feature learning

**Theoretical Claims:**

I looked at the main theorem, and the steps appeared clear and correct.

I question if the definition of a good function is somewhat arbitrary (for example, why can't I set r_1 = r_2 = 0 and ensure $\phi$ is different for each input x?)

---

> ### Author Rebuttal · Authors · 2025-04-01
>
> We thank the reviewer for their thoughtful feedback and valuable suggestions.
>
> ---
>
> **Q1**. Lack of experiments across different architectures, layers, and seeds.
>
> **A1**.  We focused on MLPs as the conventional building blocks widely used in theoretical studies.  While a full investigation across diverse architectures is beyond the scope of this theoretical paper, our core contributions establish the mathematical principles governing feature learning under different parameterizations. To improve experimental breadth, we've expanded our analysis with comprehensive eigenvalue spectrum plots across all three layers for both pre-activation and post-activation features [https://anonymous.4open.science/r/mup_rebuttal-0E0B/Rebuttal.pdf]. These visualizations show μP's advantageous properties becoming more significant in deeper layers.
>
> **Layers**: While our current plots focus on the second layer, we add results demonstrating the minimum eigenvalue trend across different depths to show that the effect persists. [https://anonymous.4open.science/r/mup_rebuttal-0E0B/Rebuttal.pdf]
>
> **Seeds**: Robustness across initializations was already considered in our experiments. As detailed in Appendix A (Experimental Details), the presented results are mean values computed over 10 independent trials using different random seeds (seeds 42-51).
>
> ---
>
> **Q2**. Questions on empirical verification of linear independence and choice of minimum eigenvalue vs. entire spectrum analysis.
>
> **A2**. Regarding the crucial empirical verification of linear independence among the feature vectors corresponding to different input data points, this is precisely achieved in our work by analyzing the minimum eigenvalue ($\lambda_{min}$) of the $N \times N$ Gram matrix computed from the centered features. This matrix, capturing the inner products between feature representations, is fundamentally related to kernel methods and the Neural Tangent Kernel (NTK).
> We compute this Gram matrix $G$. If $X$ is the $n \times N$ matrix whose columns are the centered feature vectors $h(x_i) - \bar{h}$, then the Gram matrix is $G = X^{\top} X$. (Its element $G_{ij} = (h(x_i) - \bar{h})^{\top} (h(x_j) - \bar{h})$represents the similarity between features of data points $i$and $j$). Crucially, the set of $N$ centered feature vectors $\\{h(x_i) - \bar{h}\\}\_{i=1}^N$ is linearly independent if and only if the minimum eigenvalue $\lambda{min}(G) > 0$. Linear dependence among these$N$vectors occurs precisely when $\lambda_{min}(G) = 0$. Therefore, observing $\lambda_{min}(G)$ provides a definitive and direct verification of whether the features representing different inputs maintain linear independence.
>
> While focusing on the minimum eigenvalue provides the most direct test for linear independence, we have now included comprehensive eigenvalue spectrum analysis [https://anonymous.4open.science/r/mup_rebuttal-0E0B/Rebuttal.pdf] that examines the entire distribution of eigenvalues. These plots reveal that $\mu$P maintains significantly higher eigenvalues throughout the entire spectrum compared to other parameterizations. Notably, IP exhibits catastrophic eigenvalue collapse at higher percentiles (e.g., eigenvalues dropping to $10^{-7}$ in layer 3), while $\mu$P maintains eigenvalue orders of magnitude larger across all percentiles. This full-spectrum evidence further strengthens our claims about feature diversity under $\mu$P.
>
> ---
>
> **Q3**. Definition of GOOD function arbitrary? How about $r_1=r_2=0$?:
>
> **A3**.  Regarding Assumption 4.3 / Condition 4: Here, we require that for any real numbers $r_1$ and $r_2$, the function  $(r_1 + \phi(x))(r_2 + \phi'(x))$ is not almost everywhere constant. It is worth noting that this condition involves "any real number," not the existence of some real numbers satisfying a property.
>
> The GOOD function definition, including this condition, serves as a sufficient condition that guarantees our theoretical results. For an activation function that satisfies our definition, we can show that feature learning maintains linear independence. This condition allows for a broad class of activation functions, including most commonly used ones (e.g., tanh, sigmoid).
>
> ---
>
> **Q4**. Previous works investigate covariance structure of neural networks and differentiate them from the current study i.e. they don't explicitly study cross-layer interactions.
>
> **A4**. Thank you for suggesting these valuable references. As recommended, we will add the related works "A Rainbow in Deep Network Black Boxes" and "Structured random receptive fields enable informative sensory encodings" in our revision. We will also highlight, as you noted, that these works don't explicitly study cross-layer interactions, which is one of the key differentiating aspects of our contribution.

---

### Decision · Program_Chairs · 2025-05-01

**Decision:**

Accept (poster)

**Comment:**

This paper makes a strong theoretical contribution by proving that infinite-width neural networks trained with the $\mu$-parametrization learn linearly independent features and globally converge to the minimum of the loss. It improves our understanding of feature learning dynamics in contrast to the kernel regime, using the Tensor Program framework to analyze training under SGD.

The reviewers agree that the theoretical insights are novel and technically sound. The authors well addressed concerns raised by reviewers by providing a comprehensive and technically rigorous rebuttal, including detailed derivations of the result, empirical results on full spectrum, and justifications for global convergence.

Overall, this paper offers a meaningful advance in the theoretical understanding of deep learning under the $\mu$-parametrization and is suitable for publication at ICML.